# RegMean++: Enhancing Effectiveness and Generalization of Regression Mean for Model Merging

**The-Hai Nguyen[1,*], Dang Huu-Tien[1], Takeshi Suzuki[2], and Le-Minh Nguyen[1]**
[1] *Japan Advanced Institute of Science and Technology,* [2] *Ricoh Company, Ltd.*
[*] *Correspondence to: nthehai01@jaist.ac.jp*

**Reviewed on OpenReview:** *https://openreview.net/forum?id=H5lDsSCS9i*

## Abstract

Regression Mean (RegMean), an approach that formulates model merging as a linear regression problem, aims to find the optimal weights for each linear layer in the merged model by minimizing the discrepancy in predictions between the merged and candidate models. RegMean provides a precise closed-form solution for the merging problem; therefore, it offers explainability and computational efficiency. However, RegMean merges each linear layer independently, overlooking how the features and information in earlier layers propagate through deeper layers and influence the final predictions of the merged model. Here, we introduce *RegMean++*, a simple yet effective alternative to RegMean, that explicitly incorporates both *intra-layer and cross-layer dependencies between merged models' layers* into RegMean's objective. By accounting for these dependencies, RegMean++ better captures the behaviors of the merged model. Extensive experiments demonstrate that RegMean++ consistently outperforms RegMean across diverse settings, including in-domain (ID) and out-of-domain (OOD) generalization, sequential merging, large-scale tasks, and robustness under several types of distribution shifts. Furthermore, RegMean++ achieves competitive performance across diverse settings compared to various advanced model merging methods.

## 1 Introduction

As the pretrain–finetune paradigm becomes the foundation of modern machine learning, the number of pre-trained and fine-tuned task-specific models (candidate models) is growing at an unprecedented pace. Model merging (Matena & Raffel, 2022; Wortsman et al., 2022; Ilharco et al., 2022; Jin et al., 2022; Yadav et al., 2023; Yang et al., 2024b;a; Yadav et al., 2025) aims to combine multiple candidate models into a single unified model (merged model) with multi-task capabilities, without incurring the computational overhead of traditional multi-task learning (MTL) or requiring full access to the original training data.

Regression Mean (RegMean; Jin et al. (2022)), an explainable and computationally efficient model merging method, formulates weight fusion as a closed-form regression problem that minimizes the differences between the outputs of the merged model and those of each candidate model. RegMean leverages the inner-product matrices of the input features at each candidate *linear layer*, including those within the *MLP components* and *attention heads* of transformer layers. The weights of other non-linear layers are merged by simple averaging across candidate models. RegMean offers several compelling advantages: it is model-agnostic, computation-efficient, and enables precise merging of multiple candidate models without the need for additional training. These advantages make RegMean one of the most practical merging methods.

However, a limitation of RegMean is that it **independently** applies the closed-form solution to linear layers within each transformer layer, *which may overlook how features and information are processed and propagated through layers in the merged model*, preventing it from generalizing well. These intra-layer and cross-layer dependencies are crucial for maintaining good representations that influence the final predictions of the merged model.

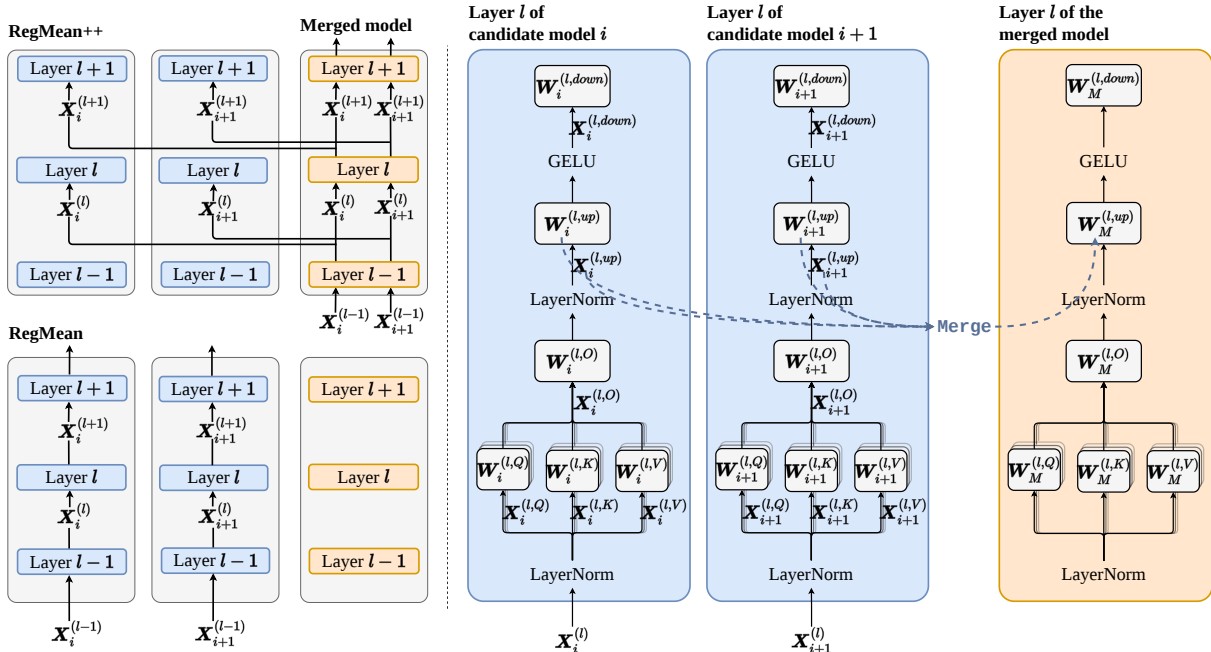

Figure 1: **Left:** Comparison of RegMean and RegMean++ for merging. RegMean++ leverages representations from the merged model, enabling more accurate alignment with its behavior. **Right:** Illustration of merging a linear layer within a transformer layer. For both methods, input features for this linear layer in the candidate models are computed by running a forward pass through the candidate models.

In this paper, we make the following contributions:

① We introduce RegMean++, a variant of RegMean applicable to both vision and language tasks. RegMean++ incorporates both intra-layer and cross-layer dependencies among layers of the merged model into the RegMean merging objective. Compared to RegMean, RegMean++ consistently improves ID performance and OOD generalization, demonstrates sustainability in sequential merging and merging with large-scale tasks. Moreover, RegMean++ shows stronger robustness under various types of distribution shifts, with reduced representation bias.

② We conduct layer-wise analysis and find that (i) merging using linear layers from only the middle and deep transformer layers preserves over 98% accuracy compared to using all layers. Conversely, earlier transformer layers appear less important, as using their linear layers results in significant accuracy degradation. (ii) Mid-depth layers consistently surpass the last layer in merging performance. This result highlights that mid-depth layers may serve as more reliable sources of meaningful features for model merging. (iii) Merging using linear layers in MLP modules consistently outperforms using those in attention heads.

③ We benchmark RegMean++ against eleven advanced model merging methods. Experimental results show that RegMean++ achieves competitive performance across diverse settings, highlighting its generalization and effectiveness relative to existing approaches.

## 2 Preliminaries

### 2.1 Notations and Problem Formulation

We denote matrices by boldface uppercase letters (*e.g.,* $\boldsymbol{X}$, $\boldsymbol{W}$, $\boldsymbol{\Lambda}$) and scalars by lowercase letters (*e.g.,* $\alpha$, $l$). For operators, we denote $||\cdot||$ the Euclidean norm and $\mathrm{tr}(\cdot)$ the trace of a matrix. Following Jin et al. (2022), we consider a *training-free merging* framework. In such a scenario, we have access to a pool of candidate models. Each candidate model is denoted by $f_i : \mathbb{R}^{N_i \times d} \to \mathbb{R}^{N_i \times |C|}$ and is fine-tuned on a task-

specific dataset $\mathcal{D}_i = \{(\boldsymbol{X}_i, \boldsymbol{Y}_i)\}$. Here, $\boldsymbol{X}_i \in \mathbb{R}^{N_i \times d}$ is a batch-input of $N_i$ samples, each with dimensionality $d$, and $\boldsymbol{Y}_i \in \mathbb{R}^{N_i \times |C|}$ is the corresponding target outputs, where $C$ is the set of classes. Our goal is to find a merging function that takes a set of $K$ candidate models $f_i$, $i \in [1..K]$, and some data points, *e.g.,* the task-specific training samples or held-out out-of-domain samples, as the inputs and returns a merged model $f_M$. We assume that all candidate models share the same architecture.

## 2.2 Regression Mean for Model Merging

Regression Mean (RegMean; Jin et al. (2022)) formulates merging as an optimization problem that minimizes the difference in outputs produced by the merged linear layer and candidate linear layers at a specific position in a transformer layer. More concretely, for each linear layer in a transformer layer $l$ of candidate model $f_i$, denoted by $\boldsymbol{W}_i^{(l)}$, given the input feature $\boldsymbol{X}_i^{(l)}$, RegMean minimizes the following regularized loss:

$$\mathcal{L}^{\text{RegMean}} = \sum_{i=1}^{K} \underbrace{||\boldsymbol{X}_i^{(l)}\boldsymbol{W}_M^{(l)} - \boldsymbol{X}_i^{(l)}\boldsymbol{W}_i^{(l)}||^2}_{\text{①}} + \sum_{i=1}^{K} \underbrace{\text{tr}\left[(\boldsymbol{W}_M^{(l)} - \boldsymbol{W}_i^{(l)})^\top \boldsymbol{\Lambda}_i^{(l)}(\boldsymbol{W}_M^{(l)} - \boldsymbol{W}_i^{(l)})\right]}_{\text{②}}, \tag{1}$$

where $\boldsymbol{W}_M^{(l)}$ is the merged linear layer's weights at the same position as $\boldsymbol{W}_i^{(l)}$ in the model $f_i$, $\boldsymbol{\Lambda}_i^{(l)} = \frac{1-\alpha}{\alpha}\text{diag}\left((\boldsymbol{X}_i^{(l)})^\top \boldsymbol{X}_i^{(l)}\right) \succeq 0$ is a regularization-strength diagonal matrix for $\boldsymbol{W}_i$, where $0 \leq \alpha \leq 1$ is a predefined scaling factor. Minimizing the loss $\mathcal{L}^{\text{RegMean}}$ in Eqn. 1 means finding $\boldsymbol{W}_M^{(l)}$ that ① approximates the behavior of all candidate linear layers while ② enforcing a regularization that keeps linear layer's weights of the merged model $\boldsymbol{W}_M^{(l)}$ close to those of the candidate $\boldsymbol{W}_i^{(l)}$. This objective describes a linear regression problem, where the inputs are $[\boldsymbol{X}_1^{(l)}, ..., \boldsymbol{X}_K^{(l)}]$ and the target outputs are $[\boldsymbol{X}_1^{(l)}\boldsymbol{W}_1^{(l)}, ..., \boldsymbol{X}_K^{(l)}\boldsymbol{W}_K^{(l)}]$. Accordingly, the closed-form solution is given as:

$$\boldsymbol{W}_M^{(l)} = \left[\sum_{i=1}^{K}\left(\widehat{\boldsymbol{G}}_i^{(l)}\right)\right]^{-1}\sum_{i=1}^{K}(\widehat{\boldsymbol{G}}_i^{(l)})\boldsymbol{W}_i^{(l)}, \tag{2}$$

where $\widehat{\boldsymbol{G}}_i^{(l)} = \alpha\boldsymbol{G}_i^{(l)} + (1-\alpha)\text{diag}(\boldsymbol{G}_i^{(l)}) = \alpha(\boldsymbol{X}_i^{(l)})^\top \boldsymbol{X}_i^{(l)} + (1-\alpha)\text{diag}((\boldsymbol{X}_i^{(l)})^\top \boldsymbol{X}_i^{(l)})$. Proof can be found in Appendix A. Other types of weights in the transformer layer are merged using averaging.

# 3 RegMean++

## 3.1 Motivation

We begin by revisiting RegMean's underlying merging mechanism. RegMean operates by **independently** applying its closed-form solution to linear layers, including those within MLPs (up-projection and down-projection matrices) and attention heads (key, query, and value matrices), across all candidate models. These components are well known for storing most of the model's learned knowledge (Meng et al., 2022), which may explain the effectiveness of RegMean.

However, deep neural networks consist of multiple non-linear components, such as GELU and LayerNorm, interleaved with linear components. Due to this non-linearity, even a small change in the input can cause a large, unpredictable shift in the output. RegMean overlooks the information flow and feature transformations that occur at both the **intra-layer level** and **cross-layer level** of the merged model. To further support this intuition, we measure the similarity between latent representations of candidate models and the merged model using Centered Kernel Alignment (CKA; Kornblith et al. (2019)) across tasks. Details of CKA are deferred to Appendix C.4. Figure 2 highlights the cross-layer (left) and intra-layer (right) CKA similarity between the RegMean model and its candidates. We observe that the similarity decreases with depth, suggesting that small changes made in early layers can propagate and disrupt representations in later layers.

We hypothesize that incorporating these dynamics, that is, intra-layer and cross-layer interactions, into RegMean's merging objective is crucial for improving the merged model's utility and generalization. Inspired by this discussion, we introduce RegMean++, a simple yet effective extension of RegMean, in Section 3.2.

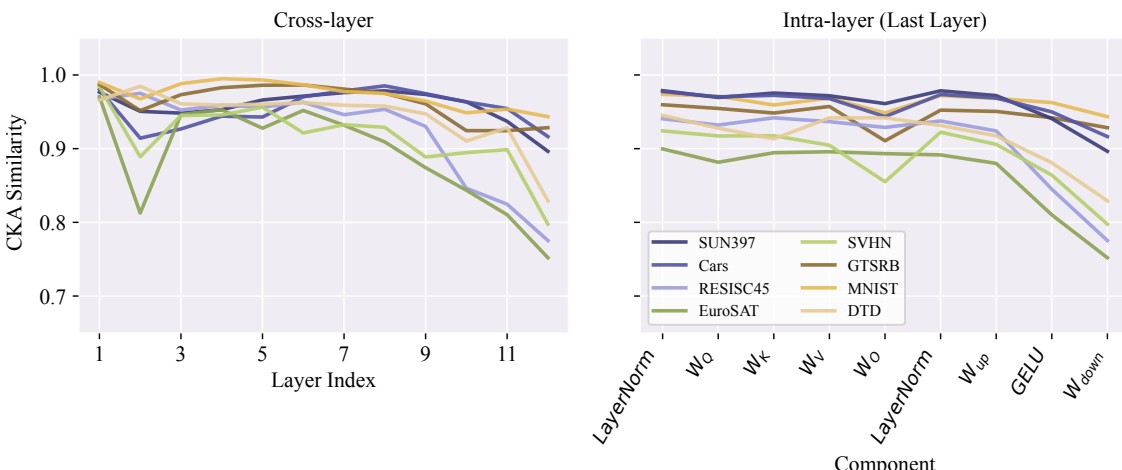

Figure 2: **Left:** Cross-layer CKA similarity between RegMean model and candidate models. **Right:** Intra-layer CKA similarity between the representations of sub-components in RegMean model and those in candidate models. Experiments are conducted on eight vision datasets for ViT-B/32.

## 3.2 RegMean++ for Model Merging

Let us fine-grain denote $\boldsymbol{X}_i^{(l,j)}$ the input feature of the $j$-th linear layer $\boldsymbol{W}_i^{(l,j)}$ in model $f_i$ at transformer layer $l$. RegMean computes the $j$-th merged linear weight $\boldsymbol{W}_M^{(l,j)}$ using Eqn. 2. In this closed-form solution, the merged weight $\boldsymbol{W}_M^{(l,j)}$ is determined by the individual weights and data statistics $\boldsymbol{G}_i^{(l,j)} = (\boldsymbol{X}_i^{(l,j)})^\top \boldsymbol{X}_i^{(l,j)}$, which captures dependencies among input features across all *candidate models*.

**Algorithm.** The key difference between Reg-Mean++ and RegMean lies in how activation $\boldsymbol{X}_i^{(l)}$ (cushion representation between transformer layers) is obtained for the current transformer layer $l$: *RegMean++ computes $\boldsymbol{X}_i^{(l)}$ based on the activation produced by the **previous merged layer** $f_M^{(l-1)}$ in the merged model, that is, $\boldsymbol{X}_i^{(l)} = f_M^{(l-1)}(\boldsymbol{X}_i^{(l-1)})$, while RegMean relies on the activation produced by the **previous candidate layer** $f_i^{(l-1)}$ in the candidate model, that is, $\boldsymbol{X}_i^{(l)} = f_i^{(l-1)}(\boldsymbol{X}_i^{(l-1)})$.* Similar to RegMean, given the $j$-th linear layer within the layer $l$, RegMean++ runs a forward pass through the candidate layer as $f_i^{(l)}(\boldsymbol{X}_i^{(l)})$ to get the input feature $\boldsymbol{X}_i^{(l,j)}$. It then computes the inner-product matrix as $\boldsymbol{G}_i^{(l,j)} = (\boldsymbol{X}_i^{(l,j)})^\top \boldsymbol{X}_i^{(l,j)}$ and reduces the non-diagonal entries by multiplying them by a scaling factor $\alpha$. Furthermore, all other non-linear parameters, such as embeddings, biases, and Layer-Norm, are merged via simple averaging. Reg-Mean++ inherits RegMean's advantages, but introduces a trade-off between model performance and computational cost. During the statistics-collection phase, RegMean++ incurs

---

**Algorithm 1** RegMean++ for Model Merging

**Require:** A pool of candidate models $f_i$ for $i \in [1...K]$, task-specific input data $\boldsymbol{X}_i$, a backbone for merging $f_M$, and a scaling factor $0 \le \alpha \le 1$.

**Ensure:** Return the merged model $f_M$.

1: **for** layer $l \in [1...L]$ **do**
2:     **for** task $i \in [1...K]$ **do**
3:         Compute $\boldsymbol{X}_i^{(l)} \leftarrow f_M^{(l-1)}(\boldsymbol{X}_i^{(l-1)})$.
4:         Run a forward pass $f_i^{(l)}(\boldsymbol{X}_i^{(l)})$.
5:     **end for**
6:     **for** linear layer $j \in [1...J]$ **do**
7:         Get input features of linear layer in $K$ candidate models: $\boldsymbol{X}_1^{(l,j)}, \boldsymbol{X}_2^{(l,j)}, ..., \boldsymbol{X}_K^{(l,j)}$.
8:         Get linear layer weights in $K$ candidate models: $\boldsymbol{W}_1^{(l,j)}, \boldsymbol{W}_2^{(l,j)}, ..., \boldsymbol{W}_K^{(l,j)}$.
9:         Compute $\boldsymbol{G}_i^{(l,j)} \leftarrow (\boldsymbol{X}_i^{(l,j)})^\top \boldsymbol{X}_i^{(l,j)}$.
10:       Compute $\widehat{\boldsymbol{G}}_i^{(l,j)} \leftarrow \alpha \boldsymbol{G}_i^{(l,j)} + (1-\alpha)\text{diag}(\boldsymbol{G}_i^{(l,j)})$.
11:       Compute $\boldsymbol{W}_M^{(l,j)}$ using Eqn. 2.
12:     **end for**
13:     Other non-linear weights are merged via averaging.
14: **end for**
15: **return** $f_M$

---

additional forward passes in the merged model to collect the inner-product matrices, yet the merging time equals that of RegMean. Our RegMean++ pseudocode is described in Algorithm 1. Comparison between RegMean and RegMean++ is shown in Figure 1.

## 4 Experiment

### 4.1 Models and Datasets

**Vision classification tasks.** Following prior works (Ilharco et al., 2022; Yang et al., 2024b; Wei et al., 2025), we evaluate the effectiveness of model merging methods on eight datasets: SUN397 (Xiao et al., 2016), Stanford Cars (Cars) (Krause et al., 2013), RESISC45 (Cheng et al., 2017), EuroSAT (Helber et al., 2019), SVHN (Netzer et al., 2011), GTSRB (Stallkamp et al., 2011), MNIST (LeCun et al., 1998), and DTD (Cimpoi et al., 2014). Following Wang et al. (2024), we employ 12 additional tasks to evaluate sustainability.

We assess the performance of merging methods on three CLIP model variants (Radford et al., 2021): ViT-B/32, ViT-B/16, and ViT-L/14. We employ off-the-shelf candidate models from Tang et al. (2025).

**Language generation tasks.** Following He et al. (2025), we evaluate merging methods on 11 datasets spanning five domains: (1) *Instruction following:* IFEval (Zhou et al., 2023), (2) *Mathematics:* GSM8K (Cobbe et al., 2021), (3) *Multilingual understanding* (on French, Spanish, German, and Russian): Multilingual MMLU, Multilingual ARC, and Multilingual Hellaswag (Lai et al., 2023), (4) *Coding:* HumanEval+ and MBPP+ (Liu et al., 2023), and (5) *Safety:* WildGuardTest (Han et al., 2024), Harm-Bench (Mazeika et al., 2024), DoAnythingNow (Shen et al., 2024), and XSTest (Röttger et al., 2024).

We assess the performance of merging methods on two Llama 3 variants (Grattafiori et al., 2024) and two Gemma 2 variants (Team et al., 2024): Llama-3.2-3B, Llama-3.1-8B, Gemma-2-2B, and Gemma-2-9B. We employ off-the-shelf candidate models from He et al. (2025).

Details of these datasets and models can be found in Appendix B.1 and B.2, respectively.

### 4.2 Comparison Methods

We compare RegMean++ against 11 model merging methods spanning three categories: (1) *Data-Free methods:* Model Soups (Wortsman et al., 2022), Task Arithmetic (Ilharco et al., 2022), TIES-Merging (Yadav et al., 2023), TSV-M (Gargiulo et al., 2025), DOGE TA (Wei et al., 2025), Iso-C and Iso-CTS (Marczak et al., 2025), (2) *Training-Free methods:* Fisher Merging (Matena & Raffel, 2022) and RegMean (Jin et al., 2022), (3) *Test-Time Adaptation:* AdaMerging (Yang et al., 2024b) and DOGE AM (Wei et al., 2025) (see Appendix B.3 for details). In addition, we consider multi-task learning as a reference performance.

### 4.3 Experimental Setup

We employ FusionBench (Tang et al., 2025) and MergeBench (He et al., 2025) to evaluate merging on vision and language tasks, respectively. For vision tasks, we report *accuracy* and *normalized accuracy*. For language tasks, we report multiple *task-specific* metrics. All metrics are calculated on the test split of each task. Hyperparameters are detailed in the corresponding subsections. See Appendix C.1, C.2, and C.3 for additional metric specifications and the full set of hyperparameters.

Due to space constraints, we present key results in the main text. We defer full experimental setups and additional results to Appendix C.5 and D.

Our code is available at `https://github.com/nthehai01/RegMean-plusplus`.

Table 1: Performance of all merging methods for ViT-B/32 measured on the 8-task benchmark. The **global best**, local best, and global runner-up are marked. See Appendix Table 10 and Table 11 for the results of ViT-B/16 and ViT-L/14.

| Method | SUN397 | Cars | RESISC45 | EuroSAT | SVHN | GTSRB | MNIST | DTD | **Avg.** |
|---|---|---|---|---|---|---|---|---|---|
| Fine-tuned | 75.0 | 78.3 | 95.2 | 99.0 | 97.3 | 98.9 | 99.6 | 79.7 | 90.3 |
| MTL | 72.3 | 76.6 | 92.2 | 97.9 | 95.5 | 97.7 | 99.3 | 77.7 | 88.6 |
| *Data-Free Methods* | | | | | | | | | |
| Model Soups | 65.4 | 62.4 | 70.6 | 75.7 | 64.5 | 55.0 | 86.3 | 50.6 | 66.3 |
| Task Arithmetic | 57.0 | 55.7 | 64.7 | 73.3 | 77.9 | 68.5 | 96.1 | 47.1 | 67.5 |
| TIES-Merging | 67.0 | 64.2 | 74.3 | 74.5 | 77.7 | 69.4 | 94.1 | 54.0 | 71.9 |
| TSV-M | 67.6 | 71.6 | 84.7 | 93.4 | 91.9 | 92.5 | 98.9 | 63.8 | 83.1 |
| DOGE TA | 67.7 | 69.9 | 81.9 | 89.8 | 86.2 | 86.8 | 98.3 | 63.8 | 80.6 |
| Iso-C | 71.0 | 73.9 | 86.0 | 89.6 | 84.8 | 90.8 | 98.2 | 65.8 | 82.5 |
| Iso-CTS | 71.1 | 74.6 | 86.6 | 89.1 | 83.4 | 90.4 | 98.1 | 68.5 | 82.7 |
| *Training-Free Methods* | | | | | | | | | |
| Fisher Merging | 67.4 | 67.6 | 75.4 | 70.5 | 76.5 | 62.2 | 87.9 | 55.3 | 70.3 |
| RegMean | 68.6 | 70.0 | 84.6 | 95.4 | 92.6 | 83.4 | 98.4 | 66.1 | 82.4 |
| **RegMean++** (Ours) | 69.3 | 70.5 | 86.7 | **96.1** | **94.1** | 90.4 | **99.0** | 68.7 | 84.4 |
| *Test-Time Adaption* | | | | | | | | | |
| Layer-wise AdaMerging | 67.8 | 71.1 | 83.9 | 92.3 | 87.8 | 93.3 | 98.2 | 66.8 | 82.6 |
| DOGE AM | 70.6 | 74.5 | **88.7** | 93.7 | 91.4 | **95.5** | 98.8 | **73.0** | **85.8** |

## 5 Results and Analysis

### 5.1 Main Results

**Merging performance.** Performance of RegMean++ and other methods on eight vision tasks for ViT-B/32 is shown in Table 1. We observe that RegMean++ consistently surpasses RegMean on all tasks, achieving an average improvement of 2.0%, and demonstrates gains of 1.2% and 0.6% on ViT-B/16 and ViT-L/14, respectively. Compared to other data-free and training-free methods, RegMean++ achieves competitive or best performance. Furthermore, despite requiring no access to test-time data or optimization, RegMean++ can rival or surpass test-time adaptation methods: outperforming Layer-wise AdaMerging (84.4% vs. 82.6%) and approaching DOGE AM (85.8%). RegMean++ achieves the best results on three specific tasks—EuroSAT (96.1%), SVHN (94.1%), and MNIST (99.0%). These results validate the advantage of leveraging the intra-layer and cross-layer dependencies.

**CKA similarity between RegMean and RegMean++.** As shown in Figure 3 and 4, RegMean++ shows higher cross-layer and intra-layer CKA similarity than RegMean in the middle and deep layers of ViT-B/32. Similar trends are observed for ViT-B/16 and ViT-L/14 in Appendix Figure 11, 12, 13, and 14.

### 5.2 Sustainability to Large-Scale Tasks

In this section, following Wang et al. (2024), we evaluate the sustainability of merging methods as the number of tasks scales up to 20. A larger number of merging tasks implies higher complexity and conflict among candidate models. Merging methods are thus expected to maintain high accuracy in this large-scale setting. Since evaluating all possible task combinations is computationally expensive, we fix the task order and add four tasks at a time. See Appendix C.5 for the experimental setting and details of the task order. Note that these experiments are conducted independently; that is, for each algorithm, the merging process is re-executed from scratch whenever new tasks are added, and performance is calculated only on the involved tasks. Performance of merging methods on vision tasks for ViT-B/32, ViT-B/16, and ViT-L/14 is visualized in Figure 5. We observe that RegMean++, along with RegMean, TSV-M, Iso-C, Iso-CTS, AdaMerging, and DOGE AM, demonstrates strong sustainability as the number of tasks increases. In contrast, Model Soups, Task Arithmetic, TIES-Merging, Fisher Merging, and DOGE TA exhibit a noticeable decline in accuracy.

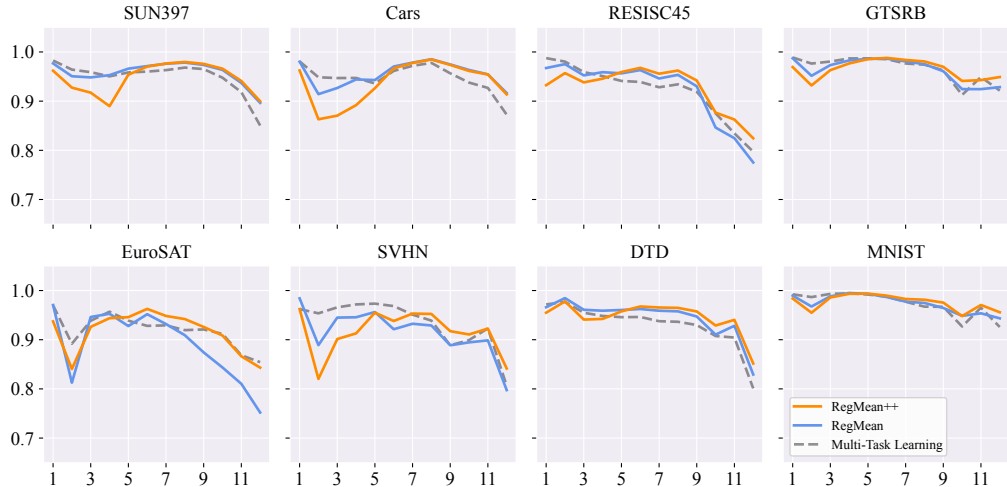

Figure 3: Cross-layer CKA similarity between the merged model and the candidate models for ViT-B/32 on eight vision tasks. We include the multi-task learning model as a reference.

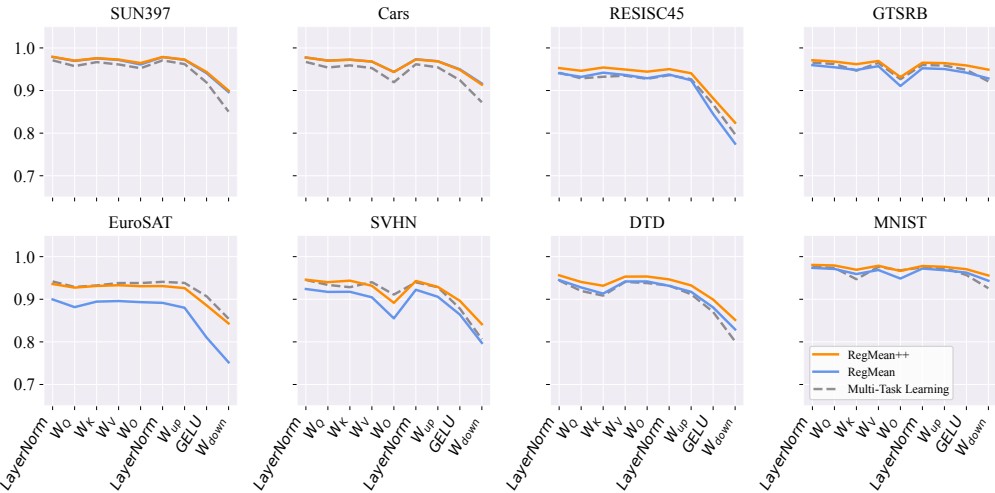

Figure 4: Intra-layer CKA similarity between the merged model and the candidate models at the last layer of ViT-B/32 on eight vision tasks. We include the multi-task learning model as a reference.

## 5.3 Sequential Merging

Merging from scratch whenever new candidate models arrive is computationally expensive and infeasible. Unlike one-time large-scale merging experimented in Section 5.2, sequential merging is a practical scenario in which new tasks arrive over time. In this setting, we evaluate the performance of RegMean and RegMean++ by merging the first four candidates in a predefined task sequence, then progressively merging the resulting model with the next four candidates, repeating this process until all 20 tasks are merged.

Figure 6 presents the performance of merging methods for all three vision models, averaged over five different task sequences. Compared to RegMean, RegMean++ demonstrates greater improvements as more candidate models are merged, especially for ViT-B/32 and ViT-B/16, where the performance gaps become more apparent as the number of merged tasks increases. Further analyses indicate that RegMean++ better accommodates new tasks while exhibiting reduced forgetting of early tasks (see Appendix D.2 for details). Compared with other merging methods, RegMean++ achieves superior performance after 8, 12, 16, and 20 tasks have been merged.

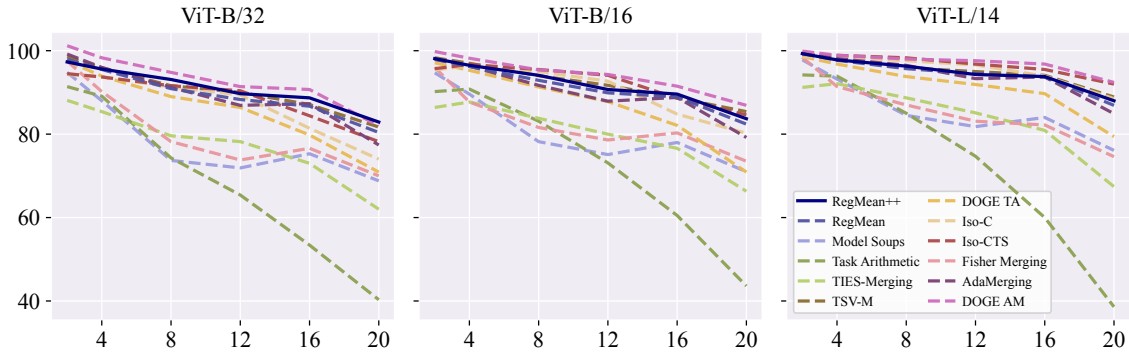

Figure 5: Average normalized accuracy of all merging methods for ViT-B/32, ViT-B/16, and ViT-L/14; evaluated on different numbers of tasks (up to 20 tasks).

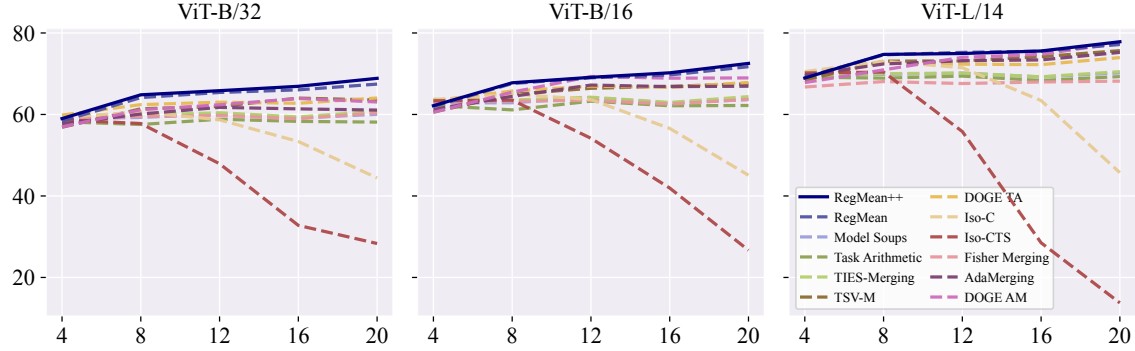

Figure 6: Sequential merging performance for ViT-B/32, ViT-B/16, and ViT-L/14. Results show the mean average accuracy on all 20 tasks across five different task sequences. **RegMean++ mitigates catastrophic forgetting better than current advanced merging methods as the sequence of tasks grows.**

## 5.4 Out-of-Domain Generalization

In this section, we evaluate the OOD generalization of merging methods. We randomly select two of the eight vision tasks to serve as OOD tasks, while the remaining six are used for merging. Table 2 reports the ID and OOD accuracy of all merging methods evaluated on three ViT models. Each result is the mean average accuracy over five runs. Across all models, RegMean++ consistently achieves strong ID performance, outperforming RegMean by margins of +1.6, +0.9, and +0.4 points for ViT-B/32, ViT-B/16, and ViT-L/14, respectively. RegMean++ slightly outperforms RegMean on OOD tasks for all three models. Notably, TIES-Merging achieves the best OOD accuracy on ViT-B/32 (51.7%), ViT-B/16 (59.9%), and ViT-L/14 (68.3%), but shows low ID performance, highlighting *a trade-off between*

Table 2: ID and OOD performance of ViT-B/32, B/16, and L/14. We report the mean of the average accuracy over five runs. The **global best**, local best, and global runner-up are marked.

| Method | ViT-B/32 | | ViT-B/16 | | ViT-L/14 | |
|---|---|---|---|---|---|---|
| | ID | OOD | ID | OOD | ID | OOD |
| *Data-Free Methods* | | | | | | |
| Model Soups | 70.2 | 50.6 | 75.4 | 58.9 | 82.7 | 67.9 |
| Task Arithmetic | 73.7 | 42.6 | 80.2 | 54.1 | 84.7 | 61.8 |
| TIES-Merging | 73.0 | **51.7** | 77.7 | **59.9** | 84.7 | **68.3** |
| TSV-M | 84.8 | 45.7 | 88.2 | 54.8 | 91.3 | 62.1 |
| DOGE TA | 83.1 | 49.4 | 86.5 | 56.0 | 89.8 | 65.8 |
| Iso-C | 83.1 | 49.3 | 87.8 | 58.2 | 92.4 | 65.9 |
| Iso-CTS | 82.8 | 48.6 | 88.1 | 58.2 | 92.8 | 65.2 |
| *Training-Free Methods* | | | | | | |
| Fisher Merging | 74.3 | 51.0 | 78.4 | 58.0 | 83.7 | 64.2 |
| RegMean | 84.6 | 48.3 | 88.0 | 56.5 | 91.5 | 64.5 |
| **RegMean++** (Ours) | 86.2 | 48.9 | 88.9 | 56.7 | 91.9 | 64.7 |
| *Test-Time Adaptation* | | | | | | |
| Layer-wise AdaMerging | 84.5 | 48.9 | 87.2 | 55.5 | 91.8 | 65.3 |
| DOGE AM | **87.4** | 47.7 | **89.6** | 54.9 | **93.0** | 63.4 |

Table 3: Performance of merging methods of ViT-B/32 on corrupted test data. **global best**, local best, and global runner-up are marked.

| Method | Clean Test Set | Corrupted Test Set | | | | | | | |
| | | Motion | Impulse | Gaussian | Pixelate | Spatter | Contrast | JPEG | **Avg.** |
|---|---|---|---|---|---|---|---|---|---|
| | | *Data-Free Methods* | | | | | | | |
| Model Soups | 76.0 | 64.6 | 56.9 | 58.1 | 28.5 | 61.3 | 64.7 | 66.3 | 57.2 |
| Task Arithmetic | 77.5 | 65.9 | 58.9 | 59.6 | 29.7 | 63.5 | 66.0 | 67.8 | 58.8 |
| TIES-Merging | 73.3 | 63.2 | 54.5 | 56.2 | 28.1 | 57.7 | 63.8 | 64.4 | 55.4 |
| TSV-M | 88.3 | 78.9 | 69.9 | 69.1 | 37.8 | 75.4 | 77.2 | 80.2 | 69.8 |
| DOGE TA | 86.1 | 77.3 | 66.0 | 66.2 | 37.5 | 71.5 | 76.1 | 77.7 | 67.5 |
| Iso-C | 84.8 | 75.9 | 62.2 | 63.9 | 35.0 | 69.2 | 76.5 | 75.3 | 65.4 |
| Iso-CTS | 84.9 | 75.7 | 61.9 | 63.5 | 34.2 | 69.2 | 76.5 | 75.4 | 65.2 |
| | | *Training-Free Methods* | | | | | | | |
| Fisher Merging | 79.1 | 67.0 | 59.8 | 60.7 | 29.3 | 64.9 | 67.9 | 69.0 | 59.8 |
| RegMean | 89.1 | 79.7 | 69.1 | 67.3 | 37.1 | 75.2 | 78.0 | 80.9 | 69.6 |
| **RegMean++** (Ours) | 89.7 | 81.8 | **70.0** | 68.4 | 37.9 | **75.9** | 79.6 | 82.8 | 70.9 |
| | | *Test-Time Adaptation* | | | | | | | |
| Layer-wise AdaMerging | 88.4 | 81.3 | 69.0 | 71.2 | 41.3 | 74.5 | 80.2 | 80.1 | 71.1 |
| DOGE AM | **90.9** | **85.0** | 65.7 | **73.4** | **44.2** | 75.4 | **83.8** | **84.2** | **73.1** |

*ID and OOD generalization in merging.* Overall, RegMean++ offers a good trade-off between ID and OOD generalization across all models, slightly outperforming RegMean. RegMean++ demonstrates competitive performance without requiring access to test data, unlike test-time adaptation methods.

## 5.5 Robustness Against Distribution Shifts

We first note that the notion of distribution shift is broad and can appear in many forms, such as covariance shift, label shift, concept shift, class-conditional shift, etc. Here, following Hendrycks & Dietterich (2019); Tang et al. (2024); Yang et al. (2024b), we evaluate the robustness of merging methods on vision test data by employing seven types of noises (covariance shift), including Motion Blur, Impulse Noise, Gaussian Noise, Pixelate, Spatter, Contrast, and JPEG Compression. These noises are introduced into four datasets: Cars, EuroSAT, RESISC45, and GTSRB. Table 3 shows the average accuracy of merging methods for ViT-B/32 under corrupted test data. We observe that RegMean++ achieves superior performance, surpassing all training-free and data-free methods on both clean and corrupted test sets. A similar trend is observed for ViT-B/16 and ViT-L/14 in Appendix Tables 12 and 13, respectively. These results highlight that RegMean++ not only performs effectively on ID and OOD tasks but is also robust to distribution shift.

## 5.6 Effects of Merging in Different Spaces

One might ask: (1) Which component—attention heads or MLPs—contributes more effectively to merging performance? (2) How does the choice of transformer layers influence the merging effectiveness?

In this section, we measure the effects of merging in different spaces (*i.e.,* different layers and components) for RegMean++ and RegMean. We conduct the following empirical experiments: (1) *Region-specific merging:* all linear layers from a specific set of transformer layers grouped by position in the model

Table 4: Region-specific and component-specific merging performance of RegMean++ and RegMean for ViT-B/32, ViT-B/16, and ViT-L/14 across eight tasks. See Appendix D.4 for other methods.

| Components | ViT-B/32 | | ViT-B/16 | | ViT-L/14 | |
| | RegMean | RegMean++ | RegMean | RegMean++ | RegMean | RegMean++ |
|---|---|---|---|---|---|---|
| All | 82.4 | 84.4 | 86.0 | 87.2 | 90.4 | 91.0 |
| Early | 67.1 | 67.8 | 73.9 | 74.8 | 81.2 | 81.5 |
| Middle | 74.0 | **77.2** | **78.9** | **81.0** | 85.6 | **86.6** |
| Deep | **74.4** | 75.7 | 78.7 | 79.5 | **85.8** | 86.4 |
| Middle & Deep | 80.7 | 83.5 | 84.4 | 86.5 | 89.4 | 90.5 |
| Attention heads | 72.9 | 76.0 | 78.0 | 80.1 | 85.0 | 86.5 |
| MLPs | **78.8** | **81.4** | **82.7** | **84.5** | **88.1** | **88.9** |

are used for merging and comparing performance across three configurations: early layers $(1, 2, 3, 4)$, middle layers $(5, 6, 7, 8)$, and deep layers $(9, 10, 11, 12)$. (2) *Layer-wise merging:* all linear layers from a specific transformer layer are used for merging. (3) *Component-specific merging:* all linear layers in MLP components or attention heads are used for merging. Note that in these experiments, only the selected linear layers are merged using the respective merging methods, while simple averaging is applied to the other linear layers. Results on vision tasks are shown in Table 4 and Figure 7.

**Merging using middle and deep layers preserves overall performance.** For all models and merging methods, using the middle and deep layers (layers 5-12 for ViT-B/32 and ViT-B/16, and layers 8-24 for ViT-L/14) achieves high performance, closely matching that of using all layers. For example, RegMean++ achieves 98%, 99%, and 99% of the full-layer accuracy for ViT-B/32 (83.5/84.4), ViT-B/16 (86.5/87.2), and ViT-L/14 (90.5/91.0). When merging is performed using only the middle or deep layers, performance remains over 90% of the full-layer. In contrast, early layers contribute less, with a notable drop in accuracy.

**MLP module merging outperforms attention head merging.** Component-specific merging analysis shows that merging using linear layers from the MLP modules yields higher accuracy than that from attention heads across all models. This implies that MLPs may contain richer semantic representations for merging. This observation aligns with previous findings (Geva et al., 2021; Meng et al., 2022; Chen et al., 2025), which have shown that MLPs serve as dictionaries of factual and task-relevant knowledge in transformer models.

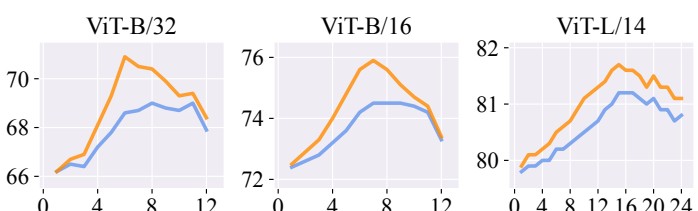

Figure 7: Layer-wise merging performance of RegMean++ (dark orange) and RegMean (cornflower blue) for ViT-B/32, ViT-B/16, and ViT-L/14. Results show average accuracy across eight tasks. See Appendix D.4 for other methods.

**Intermediate (middle and deep) layers surpass the last layer in merging performance.** Figure 7 shows that middle layers consistently surpass deep layers and the last layer in merging performance. This suggests that deep layers and the last layer may be specialized for task-specific, while middle layers serve as sources of more meaningful features beneficial for merging.

## 5.7 Effects of Data Characteristics

Data plays a central role in the training-free merging framework. However, full access to training datasets is often restricted in practice due to privacy concerns. In this section, we investigate the effects of data characteristics on merging performance for vision tasks under three settings: (1) *Number of samples:* we randomly select samples from the training set of each task; (2) *Class imbalance:* we randomly select samples from a random class in the training set of each task; and (3) *OOD samples:* we randomly select samples from the ImageNet database (Deng et al., 2009), which serves as an OOD dataset for all tasks.

**Effects of the number of ID samples.** Figure 8 shows that merging performance improves modestly as the number of ID samples increases. This implies that the effectiveness of merging is influenced more by the quality of the selected samples than by their quantity. Even a small number of ID samples can achieve near-optimal performance.

**Effects of class imbalance.** As shown in Table 5, RegMean++ performs best in this setting. However, Fisher Merging remains relatively stable with the smallest accuracy drop compared to merging in an ideal scenario, *i.e.,* class balance.

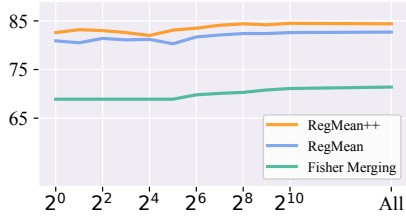

Figure 8: Impact of ID samples on training-free ViT-B/32 merging. See Appendix Figure 9 for ViT-B/16 and ViT-L/14.

**Effects of OOD samples.** Table 5 indicates that when OOD samples are used for merging, Fisher Merging demonstrates a stable performance. In contrast, RegMean and RegMean++ exhibit reduced accuracy, particularly for ViT-B/32 (*e.g.,* RegMean++ drops from 84.4% to 65.5%). This reflects a limitation of regression-based methods when the merging data distribution is misaligned with the task domains.

Table 5: Accuracy of Fisher Merging (Fisher), RegMean (RM), and RegMean++ (RM++) for ViT-B/32, ViT-B/16, and ViT-L/14 on three scenarios: random ID samples, ID class imbalance, and OOD samples. We report average accuracy across eight tasks.

| Characteristics | *ViT-B/32* | | | *ViT-B/16* | | | *ViT-L/14* | | |
|---|---|---|---|---|---|---|---|---|---|
| | Fisher | RM | RM++ | Fisher | RM | RM++ | Fisher | RM | RM++ |
| ID (random) | 70.3 | 82.4 | 84.4 | 75.6 | 86.0 | 87.2 | 82.4 | 90.4 | 91.0 |
| Class Imbalance | 69.2 | 75.5 | 76.3 | 74.5 | 80.6 | 81.6 | 77.3 | 86.0 | 86.4 |
| OOD samples | 68.5 | 66.5 | 65.5 | 73.9 | 70.2 | 71.9 | 80.6 | 78.9 | 79.6 |

### 5.8 Performance of RegMean++ on Language Tasks

Performance of RegMean and RegMean++ on language tasks for two Llama 3 variants is shown in Table 6, where each entry represents the average results across tasks in that domain. We find that the effectiveness of RegMean++ over RegMean depends on the scale of the base model. On Llama-3.1-8B, RegMean++ outperforms RegMean, where the averaged score increases from 43.5 to 45.7. It demonstrates substantial improvements in the mathematics domain (65.8 vs. 55.1) and the safety domain (46.3 vs. 33.6). Conversely, on Llama-3.2-3B, RegMean++ underperforms RegMean across all do-

Table 6: Performance of merging methods on language tasks with two Llama 3 variants.

| Method | Instruction following | Math | Multilingual | Coding | Safety | **Avg.** |
|---|---|---|---|---|---|---|
| | *Llama-3.2-3B* | | | | | |
| Model Soups | 8.7 | 36.2 | 47.8 | 37.1 | 36.5 | 33.3 |
| Task Arithmetic | 19.8 | 37.1 | **48.1** | 40.1 | **43.6** | 37.7 |
| TIES-Merging | **22.0** | **42.8** | **48.1** | **40.5** | 41.2 | **38.9** |
| RegMean | 8.3 | 35.5 | 47.3 | 39.2 | 39.8 | 34.0 |
| **RegMean++** (Ours) | 5.9 | 32.1 | 47.1 | 38.1 | 36.3 | 31.9 |
| | *Llama-3.1-8B* | | | | | |
| Model Soups | 13.9 | **67.9** | 54.4 | 50.2 | 56.2 | 48.5 |
| Task Arithmetic | 8.7 | 62.2 | 54.8 | 47.5 | 53.5 | 45.3 |
| TIES-Merging | 13.1 | 67.1 | **54.9** | 49.2 | **59.5** | **48.8** |
| RegMean | **25.9** | 55.1 | 51.4 | 51.5 | 33.6 | 43.5 |
| **RegMean++** (Ours) | 11.1 | 65.8 | 53.1 | **52.3** | 46.3 | 45.7 |

mains, where the averaged score decreases from 34.0 to 31.9. In both cases, RegMean++ underperforms RegMean on the instruction-following domain, showing a notable drop on Llama-3.1-8B (11.1 vs 25.9). Nevertheless, TIES-Merging achieves the highest overall merging performance for both base models while being more efficient (see Appendix Section D.8). Model Soups and Task Arithmetic demonstrate comparable or even higher merging performance than RegMean and RegMean++. Corresponding results for the two Gemma 2 variants are deferred to Appendix D.7.

## 6 Related Work

Model Soups (Wortsman et al., 2022; Choshen et al., 2022) is a well-known approach that performs merging by simply averaging the parameters of all candidate models. It has been used to enhance distribution robustness (Wortsman et al., 2022), and to create merged models with multi-task or multi-modality capabilities (Ilharco et al., 2022; Sung et al., 2023; Yadav et al., 2023).

Task Arithmetic (Ilharco et al., 2022) introduces the *"task vector"* concept, which quantifies the task-specific knowledge of a candidate model by measuring the difference between the candidate model parameters and the base model parameters. However, Yadav et al. (2023) point out that Task Arithmetic suffers from conflicts when different task vectors update the same parameters in opposite directions. To address this, TIES-Merging is proposed to first remove low-magnitude (noisy) parameters, then resolve sign conflicts, and finally aggregate only the non-conflicting parameters. DARE (Yu et al., 2023) pre-processes each task vector independently by applying a Bernoulli mask to randomly zero out its parameters, effectively performing dropout-like pruning before merging. TSV-M (Gargiulo et al., 2025) extracts task singular vectors via

singular value decomposition (SVD) on layer-wise weight updates, selects top-$k$ components for a low-rank representation, and applies whitening to decorrelate subspaces before merging. Marczak et al. (2025) argue that effective merging depends on how well the updates span each task's principal subspaces. To address this, Iso-C is proposed, which sums task-specific updates, performs SVD, and replaces the singular values with their average to create an isotropic spectrum. Iso-CTS, a variant of Iso-C, adds the top singular directions from each task's residual, orthogonalizes them, and then isotropically scales the result.

Inspired by test-time adaptation schemes (Wang et al., 2020; Liang et al., 2025), AdaMerging (Yang et al., 2024b) adaptively learns the merging coefficient for each layer of each task vector by minimizing entropy on unlabeled test data, using it as a surrogate objective to refine the merged model's performance across multiple tasks. DOGE AM (Wei et al., 2025) applies adaptive projective gradient descent at test time by jointly tuning a small modification vector and layer-wise merging coefficients to minimize prediction entropy on unlabeled inputs. DOGE AM projects updates orthogonally onto a shared subspace to resolve task conflicts without harming shared knowledge.

Other approaches rely on data statistics, such as Fisher Merging (Matena & Raffel, 2022), which requires computing Fisher information matrices. Daheim et al. (2024) theoretically prove that the performance drop in weighted-averaging methods (*e.g.,* Model Soups, Task Arithmetic, and Fisher Merging) is directly connected to gradient mismatch. To address this, a gradient-matching method based on a second-order approximation is proposed, which improves the merging performance and is robust to hyperparameter choices. RegMean (Jin et al., 2022) gets rid of expensive gradient computation, formulates merging as a regression problem, and then comes up with a computationally efficient and explainable closed-form solution.

## 7    Conclusion

This paper introduces RegMean++, an extension of RegMean applicable to both vision and language tasks. RegMean++ incorporates both intra-layer and cross-layer dependencies across transformer layers of the merged model into the RegMean merging objective. RegMean++ addresses RegMean's limitations in modeling information flow and feature transformations. Extensive experiments demonstrate that RegMean++ achieves robust ID performance, improved OOD generalization, and strong scalability, outperforming or matching advanced merging methods across a wide range of settings.

**Broader Impact Statement**

This work presents an algorithmic improvement to RegMean for model merging. There are dual-use concerns when merging models that may have harmful capabilities, but these are covered by existing community guidelines and may not require a separate broader impact statement for this work alone.

**Acknowledgements**

This work was partly supported by Japan Science and Technology Agency (JST) as part of Adopting Sustainable Partnerships for Innovative Research Ecosystem (ASPIRE), Grant Number JPMJAP25B2 and JST CREST, Japan, Grant Number JPMJCR2554. This work was also partly conducted in collaboration with Ricoh Company, Ltd. The authors sincerely appreciate their insightful discussions and continuous support.

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

# Appendices

# A    Derivation of RegMean's Regularized Loss

Without loss of generality, we omit the notation transformer layer $l$ of the linear layer's weights and input features. For each linear layer of candidate model $f_i$, denoted as $\boldsymbol{W}_i$, given the input feature $\boldsymbol{X}_i$, RegMean minimizes the following regularized loss:

$$\mathcal{L}^{\text{RegMean}} = \sum_{i=1}^{K} ||\boldsymbol{X}_i \boldsymbol{W}_M - \boldsymbol{X}_i \boldsymbol{W}_i||^2 + \sum_{i=1}^{K} \text{tr}\left[ (\boldsymbol{W}_M - \boldsymbol{W}_i)^\top \boldsymbol{\Lambda}_i (\boldsymbol{W}_M - \boldsymbol{W}_i) \right],$$

where $\boldsymbol{W}_M$ is the merged linear layer's weights at the same position as $\boldsymbol{W}_i$ in the model $f_i$, $\boldsymbol{\Lambda}_i = \frac{1-\alpha}{\alpha}\text{diag}\left(\boldsymbol{X}_i^\top \boldsymbol{X}_i\right) \succeq 0$ is a regularization-strength diagonal matrix for $\boldsymbol{W}_i$, where $0 \leq \alpha \leq 1$ is a predefined scaling factor. This objective has a closed-form solution as:

$$\boldsymbol{W}_M = \left( \sum_{i=1}^{K} \widehat{\boldsymbol{G}}_i \right)^{-1} \sum_{i=1}^{K} \widehat{\boldsymbol{G}}_i \boldsymbol{W}_i,$$

where $\widehat{\boldsymbol{G}}_i = \alpha \boldsymbol{G}_i + (1-\alpha)\text{diag}(\boldsymbol{G}_i) = \alpha \boldsymbol{X}_i^\top \boldsymbol{X}_i + (1-\alpha)\text{diag}(\boldsymbol{X}_i^\top \boldsymbol{X}_i)$.

*Proof.* Take derivative of $\mathcal{L}^{\text{RegMean}}$ with respect to $\boldsymbol{W}_M$:

$$\begin{aligned}
\nabla_{\boldsymbol{W}_M} \mathcal{L}^{\text{RegMean}} &= \sum_{i=1}^{K} 2\boldsymbol{X}_i^\top (\boldsymbol{X}_i \boldsymbol{W}_M - \boldsymbol{X}_i \boldsymbol{W}_i) + \sum_{i=1}^{K} (\boldsymbol{\Lambda}_i(\boldsymbol{W}_M - \boldsymbol{W}_i) + \boldsymbol{\Lambda}_i^\top(\boldsymbol{W}_M - \boldsymbol{W}_i)) \\
&= \sum_{i=1}^{K} 2(\boldsymbol{X}_i^\top \boldsymbol{X}_i \boldsymbol{W}_M - \boldsymbol{X}_i^\top \boldsymbol{X}_i \boldsymbol{W}_i) + \sum_{i=1}^{K} 2\boldsymbol{\Lambda}_i(\boldsymbol{W}_M - \boldsymbol{W}_i) \\
&= \sum_{i=1}^{K} 2(\boldsymbol{X}_i^\top \boldsymbol{X}_i + \boldsymbol{\Lambda}_i)\boldsymbol{W}_M - \sum_{i=1}^{K} 2(\boldsymbol{X}_i^\top \boldsymbol{X}_i + \boldsymbol{\Lambda}_i)\boldsymbol{W}_i.
\end{aligned}$$

We see that $\mathcal{L}^{\text{RegMean}}$ is convex. Let $\nabla_{\boldsymbol{W}_M} \mathcal{L}^{\text{RegMean}} = 0$, we can find the optimal $\boldsymbol{W}_M^*$ such that $\mathcal{L}^{\text{RegMean}}$ reaches the global minimum:

$$\boldsymbol{W}_M^* = \left[ \sum_{i=1}^{K} (\boldsymbol{X}_i^\top \boldsymbol{X}_i + \boldsymbol{\Lambda}_i) \right]^{-1} \sum_{i=1}^{K} (\boldsymbol{X}_i^\top \boldsymbol{X}_i + \boldsymbol{\Lambda}_i)\boldsymbol{W}_i. \tag{3}$$

Substitute $\boldsymbol{\Lambda}_i = \frac{1-\alpha}{\alpha}\text{diag}\left(\boldsymbol{X}_i^\top \boldsymbol{X}_i\right)$ into Eqn. 3, we have:

$$\begin{aligned}
\boldsymbol{W}_M^* &= \left[ \sum_{i=1}^{K} (\boldsymbol{X}_i^\top \boldsymbol{X}_i + \frac{1-\alpha}{\alpha}\text{diag}(\boldsymbol{X}_i^\top \boldsymbol{X}_i)) \right]^{-1} \sum_{i=1}^{K} (\boldsymbol{X}_i^\top \boldsymbol{X}_i + \frac{1-\alpha}{\alpha}\text{diag}(\boldsymbol{X}_i^\top \boldsymbol{X}_i))\boldsymbol{W}_i \\
&= \left( \sum_{i=1}^{K} \widehat{\boldsymbol{G}}_i \right)^{-1} \sum_{i=1}^{K} \widehat{\boldsymbol{G}}_i \boldsymbol{W}_i,
\end{aligned}$$

which completes the proof. □

# B  Datasets, Models, and Merging Methods

## B.1  Datasets

**Standard vision classification datasets.**  Task descriptions and statistics of datasets used for the standard 8-task image classification benchmark are described below. These datasets are publicly available `https://huggingface.co/collections/tanganke/the-eight-image-classification-tasks`.

- **SUN397** (Xiao et al., 2016) contains more than $100,000$ images of 397 categories for benchmarking scene understanding. The number of images varies across categories, but there are at least 100 images each.

- **Stanford Cars (Cars)** (Krause et al., 2013) has $16,185$ images in total of 196 types of cars and evenly split for training and testing sets.

- **RESISC45** (Cheng et al., 2017) is developed for remote sensing image scene classification. This dataset covers 45 scene classes with 700 images of size $256 \times 256$ for each.

- **EuroSAT** (Helber et al., 2019) is used for land use and land cover classification using Sentinel-2 satellite images of size $64 \times 64$, consisting of $27,000$ images covering 10 classes.

- **SVHN** (Netzer et al., 2011) is a street view house number classification benchmark, containing more than $600,000$ RGB images of 10 printed digits in size $32 \times 32$ cropped from house number plates.

- **GTSRB** (Stallkamp et al., 2011) is a German traffic sign recognition benchmark consisting of over $50,000$ images of 43 classes of traffic signs in varying light and background conditions.

- **MNIST** (LeCun et al., 1998), a well-known classical dataset for hand-written digit classification with $60,000$ training and $10,000$ testing images of size $28 \times 28$ in 10 classes of numbers.

- **DTD** (Cimpoi et al., 2014) is a collection of $5,640$ images across 47 categories of textures in the wild, annotated with human-centric attributes.

**Additional vision classification datasets.**  Besides the standard 8-task scenario, we follow the previous work and further extend our experimental scenario to 20 tasks. The new 12 tasks are listed below. These datasets are publicly available at `https://huggingface.co/collections/tanganke/image-classification-datasets`.

- **Flowers102** Nilsback & Zisserman (2008) contains 102 flower categories that are popular in the United Kingdom, with $1,020$ training and $6,149$ testing images. The images have varying poses and light conditions.

- **PCAM** (PatchCamelyon) Veeling et al. (2018) consists of more than 300M color images in size of $96 \times 96$ pixels extracted from histopathologic scans of lymph node sections. Each of them is annotated with a binary class indicating the presence of metastatic tissue.

- **FER2013** Goodfellow et al. (2013) is developed for facial expression recognition. The images are grayscale and have a size of $48 \times 48$ pixels, describing seven different kinds of emotions. The training and testing split consists of $28,709$ and $7,178$ samples, respectively.

- **OxfordIIITPet** Parkhi et al. (2012) is a 37-category pet dataset with roughly 200 images for each category, and is equally divided for both training and testing splits. The images vary in scale, pose, and lighting conditions.

- **STL10** Coates et al. (2011) is primarily built for unsupervised image recognition tasks covering 10 classes. Hence, the number of labeled images is quite small: 500 training and 800 testing images for each class. All of them are in $96 \times 96$ pixel resolution.

- **CIFAR100** Krizhevsky et al. (2009) consists of color images categorized in 100 general classes, each class contains 600 images, and each image is in size $32 \times 32$. There are $50,000$ training images and $10,000$ testing images.

- **CIFAR10** Krizhevsky et al. (2009) is similar to CIFAR100, except it has 10 classes.

- **Food101** Bossard et al. (2014) contains of 101 food categories, with $101,000$ images. For each class, 750 images are for training and 250 are for testing. Only the testing images are manually reviewed. The training images contain noise mostly from intense colors, and sometimes are mislabelled.

- **FashionMNIST** Xiao et al. (2017) is designed as a drop-in replacement benchmark for the original MNIST, thereby inheriting the same structure as MNIST.

- **EMNIST** Cohen et al. (2017) is an extended version of MNIST. EMNIST contains images of both characters and digits. We choose to use only the EMNIST Letters split, which contains around $145,000$ images evenly distributed in 26 classes of the alphabet letters.

- **KMNIST** Clanuwat et al. (2018), yet another version of MNIST, represents 10 Japanese Hiragana characters.

- **RenderedSST2** Socher et al. (2013b); Radford et al. (2019) is used for evaluating the models' capability on optical character recognition. The images are rendered from sentences in the Stanford Sentiment Treebank v2 Socher et al. (2013a), with black texts on a white background in $448 \times 448$ resolution. Each image is labeled as positive or negative based on the mood expressed in the text, and the number of images for both classes is nearly balanced. There are $6,920$ training and $1,821$ testing images.

**Language generation evaluation datasets.** We provide a detailed description of the 11 datasets used for language generation evaluation as follows.

- **IFEval** (Zhou et al., 2023) is a straightforward and easy-to-reproduce benchmark on instruction-following evaluation. It contains 541 "verifiable instructions" such as "write in more than 400 words" and "mention the keyword of AI at least 3 times". The dataset is publicly available at `https://huggingface.co/datasets/google/IFEval`.

- **GSM8K** (Cobbe et al., 2021) stands for Grade School Math 8K, which contains $8,792$ high quality grade school math problems created by human writers. These problems take between 2 and 8 steps to solve, where the solutions primarily involve performing a sequence of basic arithmetic operations ($+ - \times \div$). There are $1,319$ test problems. The dataset is publicly available at `https://huggingface.co/datasets/openai/gsm8k`.

- **Multilingual MMLU**, **Multilingual ARC**, and **Multilingual Hellaswag** (Lai et al., 2023) are the ChatGPT-translated versions from English of three corresponding datasets, *i.e,* MMLU (Hendrycks et al., 2021), ARC (Clark et al., 2018), and Hellaswag (Zellers et al., 2019). Although there are 26 languages, following (He et al., 2025), we evaluate the merged models on French, Spanish, German, and Russian. All of these datasets are organized as multiple-choice question-answering tasks, which focus on different types of knowledge. MMLU assesses the model's multi-task accuracy on a wide range of world knowledge and problem-solving ability. ARC challenges models on reasoning tasks, which comprise natural, grade-school science questions. Hellaswag provides commonsense natural language inference questions that are trivial for humans, but difficult for state-of-the-art models. These translated datasets are publicly provided as follows: Multilingual MMLU at `https://huggingface.co/datasets/alexandrainst/m_mmlu`, Multilingual ARC at `https://huggingface.co/datasets/alexandrainst/m_arc`, and Multilingual Hellaswag at `https://huggingface.co/datasets/alexandrainst/m_hellaswag`.

- **HumanEval+** and **MBPP+** (Liu et al., 2023) automatically augment the test-cases of the original HumanEval (Chen et al., 2021) and MBPP (Austin et al., 2021) datasets for code generation assessment. These benchmarks evaluate models' ability to synthesize programs from

docstrings and natural language descriptions, respectively. HumanEval+ and MBPP+ provide 80x/35x more tests than the originals, with test splits consisting of 164 and 378 programming tasks, respectively. These augmented datasets are publicly provided as follows: HumanEval+ at `https://huggingface.co/datasets/evalplus/humanevalplus` and MBPP+ at `https://huggingface.co/datasets/evalplus/mbppplus`.

- **WildGuardTest** (Han et al., 2024) is a large-scale and carefully balanced multi-task safety moderation dataset. The dataset contains $1,725$ harmful and unharmful samples covering vanilla (direct) and adversarial prompts. However, we only evaluate the merged models' ability to detect harm using 754 harmful samples. The dataset is publicly available at `https://huggingface.co/datasets/allenai/wildguardmix`.

- **HarmBench** (Mazeika et al., 2024) is a standardized evaluation framework for automated red teaming methods. HarmBench contains 400 textual behaviors, split into 320 behaviors for test and 80 behaviors for validation. These behaviors are designed to violate laws or norms, such that LLMs should not exhibit them. Each behavior is further specified with two types of categorization: semantic and functional categories. Semantic category describes the type of harmful behavior, including cybercrime, copyright violations, and generating misinformation. Functional category describes properties of behaviors, which help measure LLM's robustness. The dataset is publicly available at `https://github.com/nouhadziri/safety-eval-fork/blob/main/evaluation/tasks/generation/harmbench/harmbench_behaviors_text_test.csv`.

- **DoAnythingNow** (Shen et al., 2024) contains jailbreak prompts (spanning from December 2022 to December 2023), which are exploited by malicious users to bypass the safeguards and elicit harmful content from LLMs. These prompts are collected from four prominent platforms commonly used for prompt sharing: Reddit, Discord, websites, and open-source datasets. Following (He et al., 2025), we evaluate the merged models on a subset of 300 jailbreak prompts created from February 2023 to April 2023. The dataset is publicly available at `https://github.com/nouhadziri/safety-eval-fork/blob/main/evaluation/tasks/generation/do_anything_now/do_anything_now_jailbreak.jsonl`.

- **XSTest** (Röttger et al., 2024) evaluates whether models' safeguards are exaggerated. XSTest is inspired by a circumstance where models often struggle to balance helpfulness and harmlessness: a clearly safe prompt is even refused if it uses similar language to unsafe ones or mentions sensitive topics. XSTest contains 450 test prompts: 250 safe prompts and 200 unsafe prompts. The dataset is publicly available at `https://github.com/nouhadziri/safety-eval-fork/blob/main/evaluation/tasks/generation/xstest/exaggerated_safety.json`.

**Language generation validation datasets.** Following He et al. (2025), we employ five datasets for validation during hyperparameter tuning.

- **IFEval** (Zhou et al., 2023) is reused from the evaluation process.

- **MMLU's STEM subjects** (Hendrycks et al., 2021) include physics, computer science, mathematics, and more. Notably, these subjects are often calculation-heavy and require knowledge of empirical methods, fluid intelligence, and procedural knowledge. The dataset is publicly available at `https://huggingface.co/datasets/cais/mmlu`.

- **LAMBADA** (Paperno et al., 2016) evaluates the language understanding capabilities of models through the word prediction task on narrative passages. To accurately predict the target word, models must be able to keep track of the global information of the given passage instead of relying solely on local context. The dataset consists of $5,153$ test passages. The dataset is publicly available at `https://huggingface.co/datasets/EleutherAI/lambada_openai`.

- **CoNaLa** (Yin et al., 2018) is a benchmark for code generation task conditioned on natural language. The dataset is curated from Stack Overflow and automatically filtered to ensure high alignment

between the natural language descriptions and code snippets. To better capture user intent, the descriptions are rewritten with minimal modifications. The dataset consists of 500 test samples. The dataset is publicly available at `https://huggingface.co/datasets/neulab/conala`.

- **WildJailbreak** (Jiang et al., 2024) is a large-scale synthetic dataset designed for safety instruction tuning. It consists of vanilla harmful queries conveying unsafe requests explicitly, and adversarial harmful queries conveying unsafe requests in more convoluted and stealthy ways. The adversarial harmful queries are converted from the vanilla ones using the WildTeaming (Jiang et al., 2024) heuristic. Additionally, WildJailbreak provides vanilla and adversarial benign queries for assessing over-refusal. We evaluate the safety behaviors of the merged models using $2,000$ adversarial harmful queries. The dataset is publicly available at `https://github.com/nouhadziri/safety-eval-for k/blob/main/evaluation/tasks/generation/wildjailbreak/harmful.jsonl`.

**LLM training data.** MergeBench He et al. (2025) provides a collection of domain-specific datasets, which are used to train the candidate LLMs. We leverage these datasets to calculate the merging statistics for RegMean and RegMean++. These datasets are publicly available as follows: instruction-following domain at `https://huggingface.co/datasets/MergeBench/instruction_val`, mathematics domain at `https://huggingface.co/datasets/MergeBench/math_val`, multilingual domain at `https://huggingface.co/datasets/MergeBench/multilingual_val`, coding domain at `https://huggingface.co/datasets/Merg eBench/coding_val`, safety domain at `https://huggingface.co/datasets/MergeBench/safety_val`.

## B.2 Models

**Vision classification models.** We employ off-the-shelf fine-tuned checkpoints from the previous work (Tang et al., 2025), covering three architectures of pre-trained CLIP model (Radford et al., 2021): ViT-B/32, ViT-B/16, and ViT-L/14. We only merge the vision encoding part of these architectures, while the text encoding part is kept unchanged. The number of parameters for the vision encoding part of these architectures is 87.5M, 85.8M, and 303M, respectively. Fine-tuned checkpoints on the standard 8-task benchmark are publicly provided as follows: ViT-B/32 at `https://huggingface.co/collections/tanganke/ clip-vit-b-32-on-the-eight-image-classication-tasks`, ViT-B/16 at `https://huggingface.co/c ollections/tanganke/clip-vit-b-16-on-the-eight-image-classification-tasks`, and ViT-L/14 at `https://huggingface.co/collections/tanganke/clip-vit-l-14-on-the-eight-image-classific ation-tasks`.

**Language generation models.** We employ off-the-shelf fine-tuned checkpoints from the previous work (He et al., 2025), covering two Llama 3 variants (Grattafiori et al., 2024) and two Gemma 2 variants (Team et al., 2024): Llama-3.2-3B, Llama-3.1-8B, Gemma-2-2B, and Gemma-2-9B. For further details on these checkpoints, we refer readers to Section 3.3 of He et al. (2025). Fine-tuned checkpoints are publicly provided as follows: Llama-3.2-3B at `https://huggingface.co/collections/MergeBench/llama-32-3 b-models`, Llama-3.1-8B at `https://huggingface.co/collections/MergeBench/llama-31-8b-models`, Gemma-2-2B at `https://huggingface.co/collections/MergeBench/gemma-2-2b-models`, and Gemma-2-9B at `https://huggingface.co/collections/MergeBench/gemma-2-9b-models`.

### B.3   Merging Methods

The merging methods we employ for comparison are listed in three groups as follows:

1. *Data-Free Methods:*

   - **Model Soups** (Wortsman et al., 2022) is the most straightforward approach that simply takes the average of candidate models' parameters to produce a merged model.
   - **Task Arithmetic** (Ilharco et al., 2022) introduces a concept called "task vector", which is the difference between the fine-tuned model's parameters and the pre-trained parameters. A multi-tasking task vector is defined as the sum of those task vectors and is scaled by a coefficient before being added back to the pre-trained model's parameters to produce a merged model.
   - **TIES-Merging** (Yadav et al., 2023) proposes to trim the small values of task vectors, then resolve sign conflicts before adding back to the pre-trained parameters.
   - **TSV-M** (Gargiulo et al., 2025) compresses the task vectors using singular value decomposition (SVD) to reduce the interference between task vectors at the layer level before merging.
   - **DOGE TA** (Wei et al., 2025) is a variant of Task Arithmetic, where DOGE, an iterative algorithm minimizing the gap between the merged model and the candidates while retaining the shared knowledge, is integrated.
   - **Iso-C** and **Iso-CTS** (Marczak et al., 2025). The former applied SVD on the merged task vector to identify the directions amplified by multiple tasks, *i.e.,* common subspace. The latter further incorporates task-specific subspaces for retaining unique task features.

2. *Training-Free Methods:*

   - **Fisher Merging** (Matena & Raffel, 2022) produces the merged models by taking the weighted average of candidate models, with the weighting factors determined by the Fisher information matrices.
   - **RegMean** (Jin et al., 2022) proposes a closed-form solution for merging multiple linear layers, then applies this idea to the transformer models.

3. Test-Time Adaptation:

   - **Layer-wise AdaMerging** (Yang et al., 2024b) adaptively learns the merging coefficients introduced by Task Arithmetic in the layer-wise or task-wise manner by using unsupervised entropy minimization on unlabeled test datasets.
   - **DOGE AM** (Wei et al., 2025) is another variant of AdaMerging, where DOGE is integrated.

## C  Experimental Details

### C.1  Normalized Accuracy Metric

To avoid distortions caused by differences in value ranges, we report normalized accuracy for experiments on large-scale tasks. The normalized accuracy for each task is computed relative to the accuracy of its corresponding candidate model, then averaged over tasks as:

$$\text{Avg. Norm. Accuracy} = \frac{1}{K} \sum_{i=1}^{K} \frac{\text{acc}[f_M(\boldsymbol{X}_i)]}{\text{acc}[f_i(\boldsymbol{X}_i)]}. \tag{4}$$

### C.2  Language Tasks' Metrics

We provide the task-specific metrics for merging evaluation and validation during hyperparameter tuning on language tasks in Table 7 and Table 8, respectively.

Table 7:  Task-specific metrics for merging evaluation on language tasks.

| Domain | Dataset | Metric |
|---|---|---|
| Instruction following | IFEval (Zhou et al., 2023) | Prompt level accuracy |
| Mathematics | GSM8K (Cobbe et al., 2021) | Exact match (8-shot Chain-of-Thought) |
| Multilingual understanding (on French, Spanish, German, and Russian) | Multilingual MMLU (Lai et al., 2023) Multilingual ARC (Lai et al., 2023) Multilingual Hellaswag (Lai et al., 2023) | Accuracy Normalized accuracy Normalized accuracy |
| Coding | HumanEval+ (Liu et al., 2023) MBPP+ (Liu et al., 2023) | Pass@1 Pass@1 |
| Safety | WildGuardTest (Han et al., 2024) HarmBench (Mazeika et al., 2024) DoAnythingNow (Shen et al., 2024) XSTest (Röttger et al., 2024) | Refuse to answer Refuse to answer Refuse to answer Accuracy |

Table 8:  Task-specific metrics for merging validation on language tasks.

| Domain | Dataset | Metric |
|---|---|---|
| Instruction following | IFEval (Zhou et al., 2023) | Prompt level accuracy |
| Mathematics | MMLU's STEM subjects (Hendrycks et al., 2021) | Accuracy |
| Multilingual understanding | LAMBADA (Paperno et al., 2016) | Accuracy |
| Coding | CoNaLa (Yin et al., 2018) | BLEU score (Papineni et al., 2002) |
| Safety | WildJailbreak (Jiang et al., 2024) | Refuse to answer |

### C.3 Hyperparameters

### C.3.1 Vision Classification Tasks

Table 9: Performance of 8-task scenario merging on held-out validation sets when varying the scaling factor $\alpha$ for non-diagonal items of the inner-product matrices.

| Method | 0.10 | 0.15 | 0.20 | 0.25 | 0.30 | 0.35 | 0.40 | 0.45 | 0.50 | 0.55 | 0.60 | 0.65 | 0.70 | 0.75 | 0.80 | 0.85 | 0.90 | 0.95 | 1.00 |
|---|---|---|---|---|---|---|---|---|---|---|---|---|---|---|---|---|---|---|---|
| | | | | | | | | | *ViT-B/32* | | | | | | | | | | |
| RegMean | 76.7 | 78.4 | 79.7 | 80.7 | 81.1 | 81.9 | 82.7 | 83.3 | 83.7 | 84.2 | 84.6 | 84.9 | 85.8 | 85.9 | 86.6 | 86.9 | 87.6 | 87.6 | 3.8 |
| **RegMean++** | 78.5 | 80.0 | 81.4 | 82.1 | 83.2 | 83.6 | 84.5 | 85.0 | 85.8 | 86.2 | 86.9 | 87.4 | 87.8 | 88.0 | 88.9 | 89.4 | 89.8 | 90.1 | 5.0 |
| | | | | | | | | | *ViT-B/16* | | | | | | | | | | |
| RegMean | 80.7 | 82.0 | 82.9 | 83.6 | 84.1 | 84.9 | 85.7 | 85.8 | 86.6 | 87.0 | 87.5 | 88.0 | 88.4 | 88.6 | 89.0 | 89.1 | 89.6 | 90.3 | 4.1 |
| **RegMean++** | 82.3 | 83.8 | 84.9 | 85.6 | 86.2 | 86.9 | 87.2 | 87.8 | 88.1 | 88.5 | 88.9 | 89.4 | 90.0 | 90.2 | 90.7 | 91.0 | 91.3 | 91.6 | 4.7 |
| | | | | | | | | | *ViT-L/14* | | | | | | | | | | |
| RegMean | 87.1 | 88.0 | 88.7 | 89.4 | 89.7 | 90.2 | 90.6 | 91.1 | 91.5 | 91.7 | 92.2 | 92.4 | 92.5 | 92.8 | 93.1 | 93.4 | 93.7 | 94.0 | 4.8 |
| **RegMean++** | 88.0 | 89.2 | 89.8 | 90.6 | 91.0 | 91.6 | 91.9 | 92.2 | 92.5 | 92.6 | 93.0 | 93.3 | 93.5 | 93.7 | 94.0 | 94.5 | 94.7 | 94.8 | 4.6 |

**RegMean and RegMean++.** Both require two hyperparameters: the number of samples per task and the scaling factor $\alpha$. As illustrated in main text Figure 8 and Figure 9, increasing the number of samples for each task gives a relatively small improvement on performance across architectures. *We choose to use* 256 *samples for calculating the task's inner-product matrices* with a batch size of 32 as default. For the scaling factor $\alpha$, Table 9 shows that a higher $\alpha$ delivers better performance on the validation sets (held-out 10% of training samples). However, when $\alpha = 1.0$, *i.e.,* no scaling applied, the degradation happens and the overall accuracy is almost zeroed out. This phenomenon is consistent with the insight from Jin et al. (2022). Therefore, $\alpha = 0.95$ *is the optimal value.*

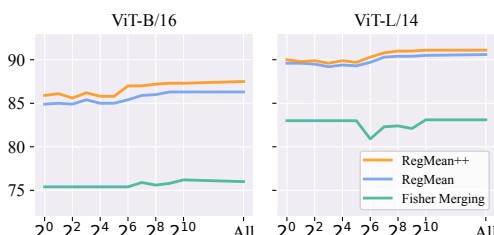

Figure 9: Effect of the amount of data needed for one candidate model when merging ViT-B/16 and ViT-L/14.

**Other methods.** We follow the suggestions on hyperparameters in the original works and set all of the hyperparameters below as default across experiments.

- For Task Arithmetic (Ilharco et al., 2022), the merging coefficient $\lambda = 0.3$ is used.

- For TIES-Merging (Yadav et al., 2023), top-20% highest-magnitude parameters are retained for each task vector, then these trimmed task vectors are merged with the merging coefficient $\lambda = 0.3$.

- For TSV-M (Gargiulo et al., 2025), the task scaling factor $\alpha = 1.0$ is used.

- For DOGE TA (Wei et al., 2025), the modification vector $\Delta$ is optimized on 400 iterations with a learning rate $1e - 4$ via Adam optimizer (Kingma & Ba, 2014), the global magnitude of merging coefficient $\eta = 0.07$, the shared subspace basis size is set as the rank of the shared subspace divided by 6, and top-30% highest-magnitude parameters are retained for each task vector.

- For Iso-C (Marczak et al., 2025), the task scaling factor $\alpha$ is set to 1.30, 1.40, and 1.50 for ViT-B/32, ViT-B/16, and ViT-L/14, respectively.

- For Iso-CTS (Marczak et al., 2025), the task scaling factor $\alpha$ is set to 1.50, 1.60, and 1.90 for ViT-B/32, ViT-B/16, and ViT-L/14, respectively; the size of the common subspace is set as its rank multiplied by 0.8.

- For Fisher Merging (Matena & Raffel, 2022), the number of samples per task is 256 and the batch size is 32 for calculating the Fisher information matrix.

- For Layer-wise AdaMerging (Yang et al., 2024b), the Adam optimizer is used with a learning rate of $1e-3$ for updating the merging coefficients on $1,000$ iterations with the batch size of 16.

- For DOGE AM (Wei et al., 2025), its hyperparameters are the same as DOGE TA and Layer-wise AdaMerging.

### C.3.2 Language Generation Tasks

**RegMean and RegMean++.** We grid search for the scaling factor $\alpha \in \{0.1, 0.3, 0.5, 0.7, 0.9\}$ and choose the ones that yield the highest averaged results across the validation datasets. For RegMean, we use $\alpha = 0.3$ for Llama-3.2-3B, $\alpha = 0.1$ for Llama-3.1-8B, $\alpha = 0.5$ for Gemma-2-2B, and $\alpha = 0.1$ for Gemma-2-9B. For RegMean++, we use $\alpha = 0.3$ for Llama-3.2-3B, $\alpha = 0.1$ for Llama-3.1-8B, $\alpha = 0.3$ for Gemma-2-2B, and $\alpha = 0.1$ for Gemma-2-9B. To compute the inner-product matrices, we use 256 samples per domain with a max sequence length of $2,048$ and batch size of 1. These samples are uniformly drawn from the candidate models' training sets (see Appendix Section B.1).

**Other methods.** Following He et al. (2025), we tune their respective hyperparameters for each method and select the configurations with the highest averaged validation results.

- For Task Arithmetic, we grid search the merging coefficient $\lambda \in \{0.1, 0.2, \ldots, 1.0\}$. We use $\lambda = 0.3$ for Llama-3.2-3B, $\lambda = 0.1$ for Llama-3.1-8B, $\lambda = 0.1$ for Gemma-2-2B, and $\lambda = 0.2$ for Gemma-2-9B.

- For TIES-Merging, we grid search the hyperparameter combinations of the merging coefficient $\lambda \in \{0.1, 0.2, \ldots, 1.0\}$ and sparsity $s = \{0.1, 0.2, 0.3\}$. We use the $(\lambda, s)$ combinations of $(0.5, 0.3)$ for Llama-3.2-3B, $(0.3, 0.2)$ for Llama-3.1-8B, $(0.1, 0.1)$ for Gemma-2-2B, and $(0.1, 0.2)$ for Gemma-2-9B.

## C.4 Details for Centered Kernel Alignment (CKA)

Centered Kernel Alignment (CKA) (Cortes et al., 2012; Kornblith et al., 2019) is a method for activation comparison within and across neural networks. CKA is a similarity index that is invariant to orthogonal transformation and isotropic scaling, but not invertible linear transformation.

Let $\boldsymbol{A} \in \mathbb{R}^{N \times d_1}$ and $\boldsymbol{B} \in \mathbb{R}^{N \times d_2}$ denote activation matrices of $d_1$ and $d_2$ neurons for $N$ samples. Instead of comparing activations of a sample from two matrices, CKA relies on similarity between every pair of samples within each activation matrix, that is, $\boldsymbol{K} = \boldsymbol{A}\boldsymbol{A}^\top$ and $\boldsymbol{L} = \boldsymbol{B}\boldsymbol{B}^\top$. Each of the Gram matrices $\boldsymbol{K}$ and $\boldsymbol{L}$ is centered to obtain $\boldsymbol{K}' = \boldsymbol{H}\boldsymbol{K}\boldsymbol{H}$ and $\boldsymbol{L}' = \boldsymbol{H}\boldsymbol{L}\boldsymbol{H}$, where $\boldsymbol{H} = \boldsymbol{I}_N - \frac{1}{N}\mathbf{1}\mathbf{1}^\top$ is the centering matrix. Hilbert-Schmidt Independence Criterion (HSIC) (Gretton et al., 2005) computes similarity between these centered similarity matrices as:

$$\text{HSIC}_0(\boldsymbol{K}, \boldsymbol{L}) = \frac{\text{vec}(\boldsymbol{K}') \cdot \text{vec}(\boldsymbol{L}')}{(N-1)^2}, \tag{5}$$

where $\text{vec}(\cdot)$ denotes the vectorization operation. Finally, CKA normalizes HSIC to produce the similarity index between 0 and 1:

$$\text{CKA}(\boldsymbol{K}, \boldsymbol{L}) = \frac{\text{HSIC}_0(\boldsymbol{K}, \boldsymbol{L})}{\sqrt{\text{HSIC}_0(\boldsymbol{K}, \boldsymbol{K})\text{HSIC}_0(\boldsymbol{L}, \boldsymbol{L})}}. \tag{6}$$

**Minibatch CKA.** Computing CKA over an entire dataset $\mathcal{D}$ requires storing all activations, which is computationally expensive. We adapt minibatch CKA (Nguyen et al., 2021) that computes CKA by averaging HISC scores over $k$ minibatches:

$$\text{CKA}_{\text{minibatch}} = \frac{\frac{1}{k}\sum_{i=1}^{k} \text{HSIC}_1(\boldsymbol{A}_i\boldsymbol{A}_i^\top, \boldsymbol{B}_i\boldsymbol{B}_i^\top)}{\sqrt{\frac{1}{k}\sum_{i=1}^{k}\text{HSIC}_1(\boldsymbol{A}_i\boldsymbol{A}_i^\top, \boldsymbol{A}_i\boldsymbol{A}_i^\top)}\sqrt{\frac{1}{k}\sum_{i=1}^{k}\text{HSIC}_1(\boldsymbol{B}_i\boldsymbol{B}_i^\top, \boldsymbol{B}_i\boldsymbol{B}_i^\top)}}, \tag{7}$$

where $\boldsymbol{A}_i \in \mathbb{R}^{n \times d_1}$ and $\boldsymbol{B}_i \in \mathbb{R}^{n \times d_2}$ are the $i$-th batch of $n$ samples. $\text{HSIC}_1$ is the unbiased esitmator of HSIC (Song et al., 2012):

$$\text{HSIC}_1(\boldsymbol{K}, \boldsymbol{L}) = \frac{1}{n(n-3)}\left(\text{tr}(\tilde{\boldsymbol{K}}\tilde{\boldsymbol{L}}) + \frac{\mathbf{1}^\top\tilde{\boldsymbol{K}}\mathbf{1}\mathbf{1}^\top\tilde{\boldsymbol{L}}\mathbf{1}}{(n-1)(n-2)} - \frac{2}{n-2}\mathbf{1}^\top\tilde{\boldsymbol{K}}\tilde{\boldsymbol{L}}\mathbf{1}\right), \tag{8}$$

where $\tilde{\boldsymbol{K}}$ and $\tilde{\boldsymbol{L}}$ are $\boldsymbol{K}$ and $\boldsymbol{L}$ with zero diagonal entries, respectively.

**Implementation.** We compute minibatch CKA using the test splits of the vision datasets. Following (Nguyen et al., 2021; Kim & Han, 2023), we iterate over each dataset 10 times. We set batch sizes of 128, 64, and 8 for ViT-B/32, ViT-B/16, and ViT-L/14, respectively.

### C.5  Details for Experimental Settings

**Order of tasks for sustainability evaluation.** We assess the merging performance by varying the number of tasks $n \in \{2, 4, 8, 12, 16, 20\}$. For every iteration $i$, we get first $n_i$ tasks from a fixed sequence of 20 tasks, perform merging, and then *evaluate the performance on the $n_i$ involved tasks only*. These experiments are conducted independently and re-executed from scratch for every iteration. The fixed sequence of 20 tasks is as follows:

- SUN397, Stanford Cars, RESISC45, EuroSAT, SVHN, GTSRB, MNIST, DTD, Flowers102, PCAM, FER2013, OxfordIIITPet, STL10, CIFAR100, CIFAR10, Food101, FashionMNIST, EMNIST, KMNIST, RenderedSST2.

**Procedure for sequential merging.** We assess the merging performance in a scenario where tasks arrive sequentially. We choose to merge four tasks at a time. Specifically, we first merge four task-specific models to obtain an initial merged model. Then, this merged model is further merged with the new four task-specific models, using the ID data for calculating its merging statistics. That is, a mixture of task-specific datasets is constructed, where the number of samples used for each task is simply defined as 256 divided by the total number of tasks involved so far. This merging process is repeated until all of 20 task-specific models are merged. Right after every merged model is obtained, we *evaluate its performance on all* 20 *tasks*.

To determine the order of merging tasks, we generate a batch of different task combinations and choose five task sequences among them such that every non-overlapping group of four tasks in a task sequence does not exist in the other sequences. The five task sequences are as follows:

- PCAM, FER2013, OxfordIIITPet, RenderedSST2, GTSRB, FashionMNIST, SUN397, CIFAR100, EuroSAT, Stanford Cars, MNIST, STL10, DTD, Flowers102, CIFAR10, Food101, KMNIST, EMNIST, SVHN, RESISC45.

- CIFAR100, SUN397, EMNIST, EuroSAT, RESISC45, Food101, Flowers102, PCAM, RenderedSST2, Stanford Cars, CIFAR10, GTSRB, MNIST, DTD, KMNIST, FashionMNIST, STL10, SVHN, OxfordIIITPet, FER2013.

- EuroSAT, RenderedSST2, SUN397, FashionMNIST, Food101, KMNIST, OxfordIIITPet, DTD, PCAM, FER2013, Flowers102, MNIST, RESISC45, Stanford Cars, CIFAR10, STL10, GTSRB, EMNIST, SVHN, CIFAR100.

- EMNIST, RESISC45, MNIST, CIFAR10, FashionMNIST, SVHN, KMNIST, STL10, GTSRB, EuroSAT, SUN397, PCAM, Flowers102, FER2013, OxfordIIITPet, Food101, DTD, RenderedSST2, Stanford Cars, CIFAR100.

- GTSRB, Stanford Cars, SUN397, FashionMNIST, CIFAR10, EMNIST, SVHN, FER2013, OxfordIIITPet, Food101, MNIST, RenderedSST2, DTD, CIFAR100, Flowers102, PCAM, KMNIST, STL10, EuroSAT, RESISC45.

**Unseen tasks for evaluating the OOD generalization ability.** We report the performance over five held-out sets for OOD tasks: {MNIST, DTD}, {SVHN, GTSRB}, {RESISC45, EuroSAT}, {SUN397, Cars}, and {Cars, RESISC45}. Meanwhile, the remaining six tasks in the standard benchmark serve as ID tasks. These sets are chosen such that the overlapping rate between OOD sets is the least.

**Example of corrupted images for evaluating the robustness.** Figure 10 demonstrates seven corrupted variants for a clean image drawn from the Stanford Cars dataset.

**ImageNet sampling (OOD sampling).** Due to its massive volume, we randomly select 256 from the first $10,000$ samples of this database. These selected samples are used as proxy task-specific data for all of the candidate models.

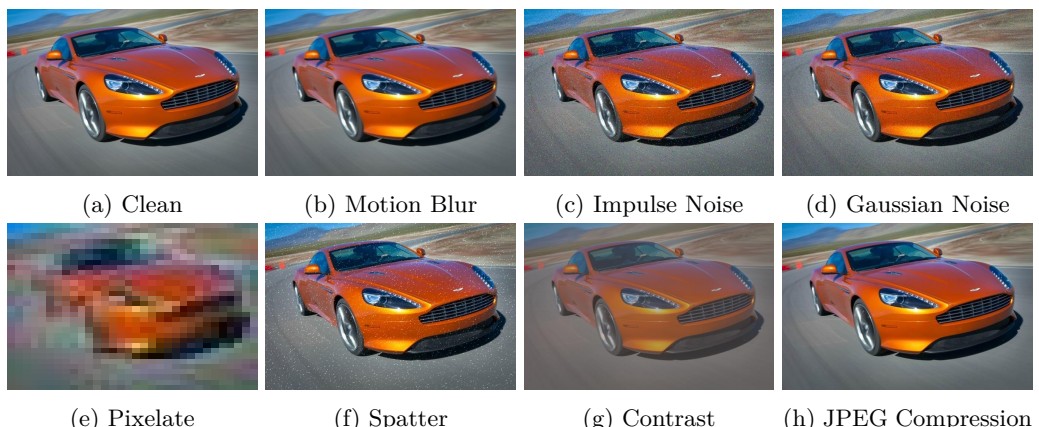

(a) Clean    (b) Motion Blur    (c) Impulse Noise    (d) Gaussian Noise

(e) Pixelate    (f) Spatter    (g) Contrast    (h) JPEG Compression

Figure 10: A visualization of a clean image and its corrupted versions with seven types of common noises (Hendrycks & Dietterich, 2019).

**Class-imbalance sampling.** We simulate a data-limited scenario where only a single class from each task-specific dataset is utilized for merging. We execute five runs for each of the training-free methods, using different classes from the set $\{0, 1, 2, 3, 4\}$ for each run. For example, the first run involves class 0; meaning that for a task dataset, at most 256 samples labeled as 0 are randomly selected to serve as data for merging.

## C.6    Hardware

All of the experiments were conducted on either an A100 GPU with 40GB memory or an A40 GPU with 48GB memory.

# D   Additional Results and Analysis

## D.1   Performance on the 8-Task Benchmark

We provide the performance comparison of RegMean++ and other methods on the 8-task benchmark for ViT-B/16 and ViT-L/14 in Table 10 and Table 11, respectively. Cross-layer and intra-layer CKA similarity results for RegMean and RegMean++ are in Figure 11, 12, 13, and 14.

Table 10:   Performance of all merging methods for ViT-B/16 measured on the 8-task benchmark. The **global best**, local best, and global runner-up are marked.

| Method | SUN397 | Cars | RESISC45 | EuroSAT | SVHN | GTSRB | MNIST | DTD | **Avg.** |
|---|---|---|---|---|---|---|---|---|---|
| *Reference Results* | | | | | | | | | |
| Fine-tuned | 78.9 | 85.9 | 96.6 | 99.0 | 97.6 | 99.0 | 99.7 | 82.3 | 92.3 |
| *Data-Free Methods* | | | | | | | | | |
| Model Soups | 68.7 | 69.0 | 75.1 | 83.3 | 75.0 | 62.6 | 93.8 | 51.2 | 72.3 |
| Task Arithmetic | 65.9 | 68.3 | 75.5 | 84.5 | 88.9 | 82.0 | 98.1 | 54.0 | 77.1 |
| TIES-Merging | 70.7 | 71.2 | 79.9 | 87.5 | 83.3 | 76.3 | 96.4 | 55.5 | 77.6 |
| TSV-M | 73.1 | 80.7 | 89.7 | 96.2 | 94.1 | 94.1 | **99.1** | 69.7 | 87.1 |
| DOGE TA | 70.8 | 77.5 | 85.9 | 95.1 | 92.7 | 91.4 | 98.8 | 65.3 | 84.7 |
| Iso-C | 75.9 | 82.9 | 92.3 | 96.3 | 91.1 | 94.5 | 98.7 | 71.2 | 87.9 |
| Iso-CTS | **76.2** | **83.8** | **92.6** | 96.0 | 90.9 | 94.7 | 98.6 | 73.7 | **88.3** |
| *Training-Free Methods* | | | | | | | | | |
| Fisher Merging | 68.5 | 69.7 | 73.6 | 96.3 | 78.8 | 73.3 | 90.4 | 54.0 | 75.6 |
| RegMean | 72.6 | 78.8 | 89.2 | 96.3 | 94.9 | 90.0 | 98.8 | 67.9 | 86.0 |
| **RegMean++** (Ours) | 72.8 | 78.9 | 89.3 | **97.3** | **96.0** | 93.0 | **99.1** | 71.0 | 87.2 |
| *Test-Time Adaption Methods* | | | | | | | | | |
| Layer-wise AdaMerging | 70.7 | 79.8 | 86.5 | 93.4 | 93.7 | 95.6 | 98.1 | 62.7 | 85.1 |
| DOGE AM | 72.6 | 82.4 | 90.5 | 94.6 | 94.8 | **96.4** | 98.6 | **75.8** | 88.2 |

Table 11:   Performance of all merging methods for ViT-L/14 measured on the 8-task benchmark. The **global best**, local best, and global runner-up are marked.

| Method | SUN397 | Cars | RESISC45 | EuroSAT | SVHN | GTSRB | MNIST | DTD | **Avg.** |
|---|---|---|---|---|---|---|---|---|---|
| *Reference Results* | | | | | | | | | |
| Fine-tuned | 82.8 | 92.9 | 97.4 | 99.2 | 97.9 | 99.2 | 99.8 | 85.5 | 94.3 |
| MTL | 79.0 | 89.3 | 94.5 | 98.4 | 96.4 | 98.1 | 99.4 | 83.7 | 92.4 |
| *Data-Free Methods* | | | | | | | | | |
| Model Soups | 72.5 | 81.5 | 82.3 | 88.5 | 81.6 | 74.0 | 96.6 | 61.8 | 79.9 |
| Task Arithmetic | 72.0 | 79.0 | 80.6 | 84.6 | 87.5 | 83.5 | 98.0 | 58.5 | 80.5 |
| TIES-Merging | 74.8 | 83.2 | 86.5 | 89.7 | 89.7 | 85.2 | 98.7 | 63.9 | 83.8 |
| TSV-M | 78.2 | 89.8 | 93.5 | 96.7 | 95.6 | 96.5 | 99.1 | 75.3 | 90.6 |
| DOGE TA | 76.6 | 87.5 | 91.3 | 96.0 | 94.4 | 93.5 | 98.9 | 71.3 | 88.7 |
| Iso-C | **80.7** | 91.5 | 95.3 | 97.2 | 95.1 | 97.8 | 99.1 | 80.3 | 92.1 |
| Iso-CTS | **80.7** | **92.2** | **95.9** | **97.5** | 95.7 | **98.4** | **99.2** | 82.1 | **92.7** |
| *Training-Free Methods* | | | | | | | | | |
| Fisher Merging | 70.9 | 78.8 | 83.0 | 94.7 | 84.9 | 94.9 | 91.1 | 61.0 | 82.4 |
| RegMean | 76.9 | 89.8 | 93.0 | **97.5** | 96.3 | 94.1 | 98.7 | 77.0 | 90.4 |
| **RegMean++** (Ours) | 77.2 | 89.6 | 92.8 | **97.5** | **96.9** | 96.3 | **99.2** | 78.4 | 91.0 |
| *Test-Time Adaption Methods* | | | | | | | | | |
| Layer-wise AdaMerging | 78.2 | 90.8 | 90.8 | 96.1 | 95.0 | 97.5 | 98.5 | 81.4 | 91.0 |
| DOGE AM | 79.6 | 91.8 | 94.2 | 96.8 | 96.3 | **98.6** | 98.9 | **83.8** | 92.5 |

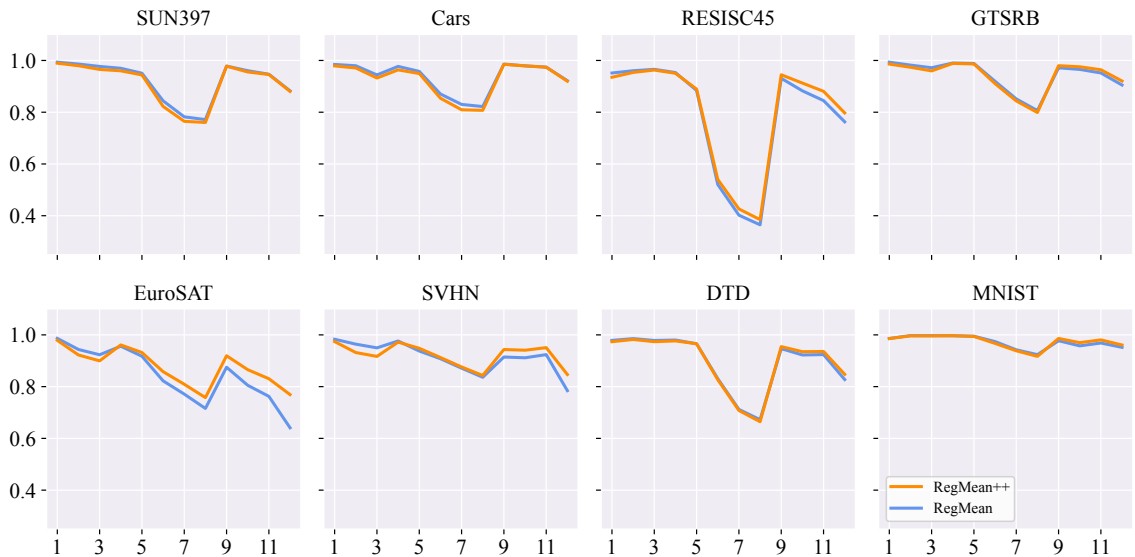

Figure 11: Cross-layer CKA similarity between the merged model and the candidate models for ViT-B/16 on eight vision tasks.

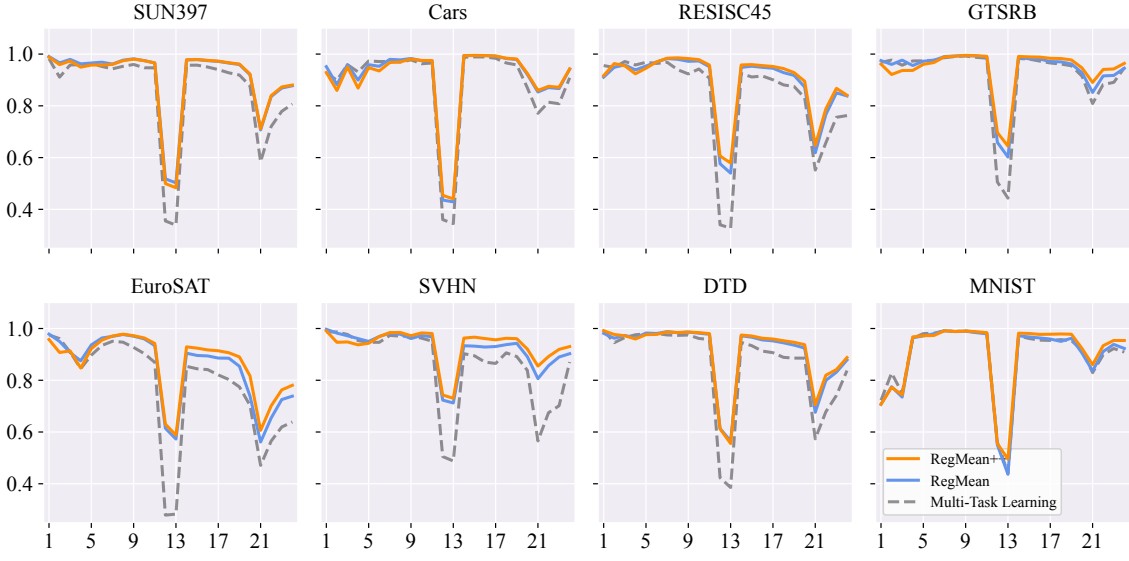

Figure 12: Cross-layer CKA similarity between the merged model and the candidate models for ViT-L/14 on eight vision tasks. We include the multi-task learning model as a reference.

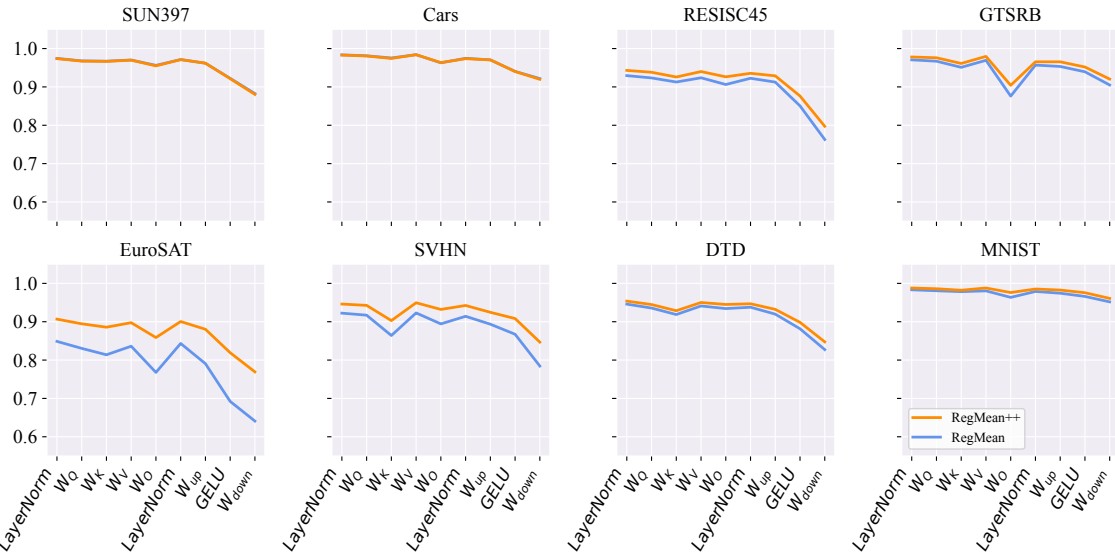

Figure 13: Intra-layer CKA similarity between the merged model and the candidate models at the last layer of ViT-B/16 on eight vision tasks.

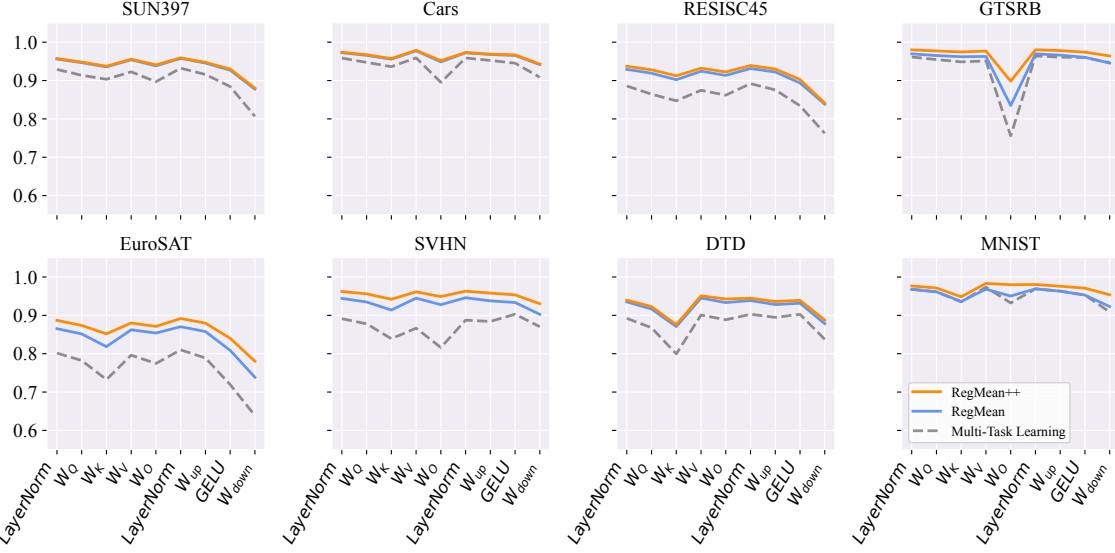

Figure 14: Intra-layer CKA similarity between the merged model and the candidate models at the last layer of ViT-L/14 on eight vision tasks. We include the multi-task learning model as a reference.

## D.2 Sequential Merging

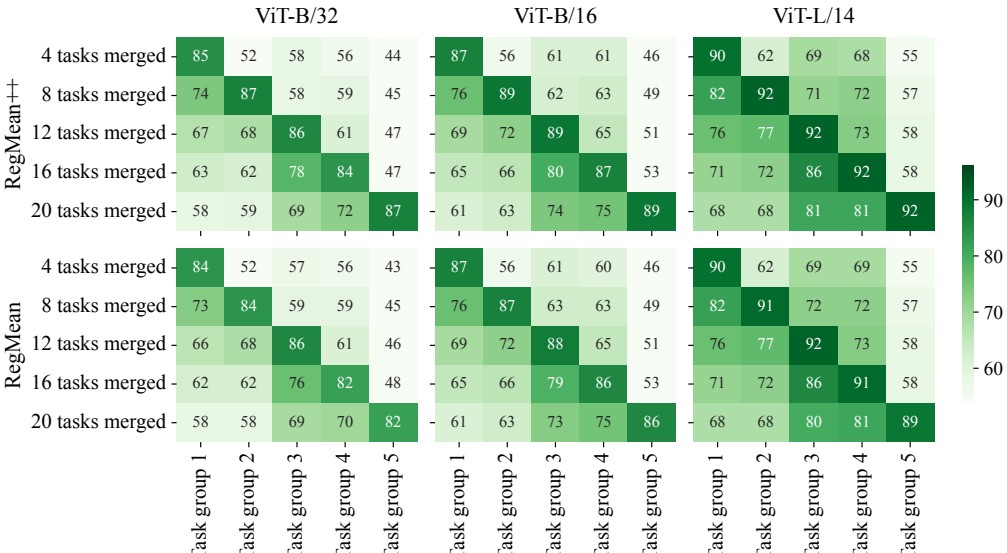

Figure 15: Details on sequential-merging performance comparison of RegMean and RegMean++ for three models. Each entry is the average accuracy on a group of four tasks, averaged over five runs.

Along with the average performance on all 20 tasks shown in main text Figure 6, we additionally provide a more fine-grained analysis of RegMean and RegMean++ in Figure 15. Each entry in these heat maps visualizes the mean of average accuracy on a non-overlapping group of four tasks across five different task sequences. Along diagonals, which correspond to groups of current merging tasks, RegMean++'s performance matches or surpasses that of RegMean. Furthermore, RegMean++ also exhibits a slightly enhanced ability to retain performance on earlier tasks, as indicated by the entries below those diagonals.

## D.3 Robustness Against Distribution Shifts

We provide the performance of RegMean++ and other methods on the robustness analysis for ViT-B/16 and ViT-L/14 in Table 12 and Table 13, respectively.

Table 12: Performance of merging methods for ViT-B/16 on corrupted test data. **global best**, local best, and global runner-up are marked.

| Method | Clean Test set | Corrupted Test set | | | | | | | |
|---|---|---|---|---|---|---|---|---|---|
| | | Motion | Impulse | Gaussian | Pixelate | Spatter | Contrast | JPEG | **Avg.** |
| *Data-Free Methods* | | | | | | | | | |
| Model Soups | 81.9 | 73.6 | 62.0 | 64.3 | 34.4 | 64.3 | 73.2 | 72.6 | 63.5 |
| Task Arithmetic | 84.0 | 75.8 | 64.1 | 65.9 | 35.4 | 66.5 | 75.2 | 74.6 | 65.3 |
| TIES-Merging | 78.4 | 69.2 | 57.5 | 59.7 | 30.3 | 60.7 | 68.7 | 68.6 | 59.2 |
| TSV-M | 92.3 | 86.0 | 74.6 | 73.3 | 42.0 | 77.5 | 84.1 | 84.8 | 74.6 |
| DOGE TA | 90.5 | 83.7 | 71.9 | 70.7 | 40.7 | 74.5 | 82.1 | 82.0 | 72.2 |
| Iso-C | 90.3 | 83.3 | 67.8 | 69.2 | 39.4 | 73.3 | 82.6 | 81.0 | 70.9 |
| Iso-CTS | 90.7 | 83.7 | 67.9 | 69.6 | 38.7 | 74.1 | 83.2 | 81.5 | 71.2 |
| *Training-Free Methods* | | | | | | | | | |
| Fisher Merging | 81.4 | 73.6 | 58.4 | 59.9 | 33.6 | 63.7 | 72.4 | 72.0 | 61.9 |
| RegMean | 92.6 | 86.2 | **75.6** | 73.3 | 41.9 | 77.7 | 84.7 | 84.8 | 74.9 |
| **RegMean++** (Ours) | **93.2** | 87.3 | 75.3 | 72.7 | 42.9 | 77.4 | 84.2 | 86.5 | 75.2 |
| *Test-Time Adaptation* | | | | | | | | | |
| Layer-wise AdaMerging | 90.9 | 84.6 | 71.7 | 73.0 | 42.9 | 75.7 | 83.3 | 83.6 | 73.5 |
| DOGE AM | 93.1 | **87.6** | 75.4 | **76.0** | **46.4** | **79.3** | **86.2** | **86.6** | **76.8** |

Table 13: Performance of merging methods for ViT-L/14 on corrupted test data. **global best**, local best, and global runner-up are marked.

| Method | Clean Test set | Corrupted Test set | | | | | | | |
|---|---|---|---|---|---|---|---|---|---|
| | | Motion | Impulse | Gaussian | Pixelate | Spatter | Contrast | JPEG | **Avg.** |
| *Data-Free Methods* | | | | | | | | | |
| Model Soups | 89.7 | 82.3 | 71.7 | 72.3 | 37.9 | 75.1 | 81.2 | 80.8 | 71.6 |
| Task Arithmetic | 90.7 | 82.8 | 73.0 | 72.7 | 37.7 | 76.6 | 81.9 | 81.4 | 72.3 |
| TIES-Merging | 87.9 | 80.9 | 69.0 | 71.4 | 37.5 | 71.9 | 80.2 | 79.1 | 70.0 |
| TSV-M | 95.5 | 89.3 | 81.4 | 80.8 | 42.4 | 86.6 | 87.1 | 88.3 | 79.4 |
| DOGE TA | 94.4 | 88.5 | 79.0 | 79.8 | 44.1 | 83.8 | 87.0 | 87.0 | 78.5 |
| Iso-C | 95.7 | 90.5 | 79.5 | 81.8 | 45.9 | 86.1 | 89.1 | 88.5 | 80.2 |
| Iso-CTS | 96.2 | 90.8 | 81.0 | 82.4 | 45.3 | 87.4 | 89.5 | 89.0 | 80.8 |
| *Training-Free Methods* | | | | | | | | | |
| Fisher Merging | 89.2 | 81.9 | 73.7 | 72.5 | 40.2 | 77.6 | 80.8 | 81.0 | 72.5 |
| RegMean | 95.9 | 89.2 | 83.5 | 80.9 | 40.9 | 87.2 | 87.5 | 88.7 | 79.7 |
| **RegMean++** (Ours) | 96.1 | 89.8 | **83.7** | 80.4 | 41.5 | 87.4 | 87.5 | 89.4 | 79.9 |
| *Test-Time Adaptation* | | | | | | | | | |
| Layer-wise AdaMerging | 95.3 | 91.2 | 78.9 | 84.2 | **50.8** | **88.0** | 89.8 | 90.3 | 81.9 |
| DOGE AM | **96.4** | **91.6** | 80.8 | **85.5** | 50.6 | 86.6 | **90.8** | **91.3** | **82.5** |

### D.4 Merging in Different Spaces for Data-Free Methods

**Merging using middle and deep layers preserves overall performance or even outperforms all layers.** The results are reported in Table 14. For Task Arithmetic, TIES-Merging, TSV-M, and DOGE TA using both middle and deep layers achieves highly competitive performance, or even surpasses that of using all layers. The most noticeable performance differences can be observed on Task Arithmetic, where it consistently improves over full-layer merging by 6.6, 2.0, and 3.5 points for ViT-B/32, ViT-B/16, and ViT-L/14, respectively. Meanwhile, merging both middle and deep layers preserves from 98% to 99% performance for both Iso-C and Iso-CTS. Additionally, applying merging to only deep layers retains more than 92% overall performance, and is better than merging using middle or early layers. This trend can also be observed in layer-wise merging in Figure 16.

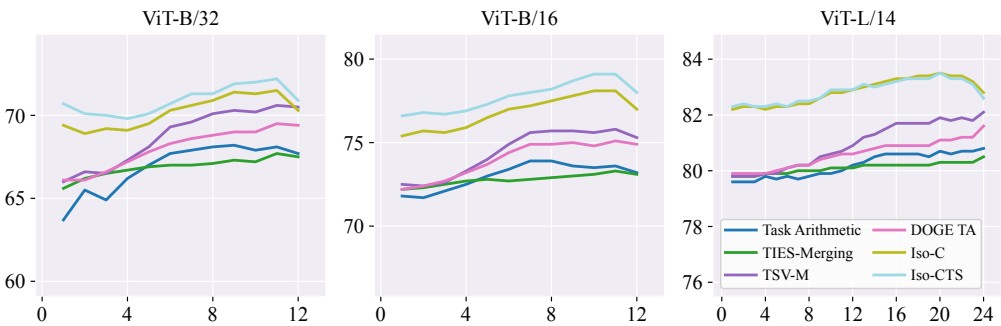

Figure 16: Layer-wise merging performance of data-free methods for ViT-B/32. ViT-B/16, and ViT-L/14. Results show average accuracy across eight tasks.

**MLP module linear layer merging is not always the best.** Component-specific merging analysis for data-free methods shows that using linear layers of the MLP modules still yields better performance than that of the attention heads, except for Task Arithmetic, as indicated in Table 14.

Table 14: Region-specific and component-specific merging performance of data-free methods for ViT-B/32, ViT-B/16, and ViT-L/14, where Task Arithmetic is denoted as TA and TIES-Merging is denoted as TIES. Results show average accuracy across eight tasks.

| Components | ViT-B/32 | | | | | | ViT-B/16 | | | | | | ViT-L/14 | | | | | |
|---|---|---|---|---|---|---|---|---|---|---|---|---|---|---|---|---|---|---|
| | TA | TIES | TSV-M | DOGE TA | Iso-C | Iso-CTS | TA | TIES | TSV-M | DOGE TA | Iso-C | Iso-CTS | TA | TIES | TSV-M | DOGE TA | Iso-C | Iso-CTS |
| All | 67.5 | 71.9 | 83.1 | 80.6 | 82.5 | 82.7 | 77.1 | 77.6 | 87.1 | 84.7 | 87.9 | 88.3 | 80.5 | 83.8 | 90.6 | 88.7 | 92.1 | 92.7 |
| Early | 60.5 | 65.8 | 66.6 | 66.5 | 70.7 | 71.7 | 70.7 | 72.7 | 73.2 | 72.8 | 77.3 | 78.3 | 76.6 | 79.8 | 80.1 | 80.4 | 83.9 | 84.8 |
| Middle | 70.1 | 69.0 | 74.9 | 72.8 | 74.7 | 75.0 | 76.4 | 74.1 | 80.2 | 79.0 | 81.5 | 81.6 | 80.9 | 81.7 | 85.9 | 84.4 | 87.3 | 87.9 |
| Deep | 72.1 | 70.6 | 78.5 | 76.3 | 77.0 | 76.9 | 76.4 | 75.5 | 82.6 | 81.1 | 83.4 | 83.9 | 84.1 | 82.7 | 88.1 | 86.4 | 88.8 | 88.7 |
| Middle & Deep | 74.1 | 73.1 | 83.4 | 81.0 | 81.6 | 81.0 | 79.1 | 77.3 | 87.0 | 84.9 | 87.0 | 87.0 | 84.0 | 84.2 | 90.9 | 88.6 | 91.4 | 91.8 |
| Attention heads | 71.1 | 69.2 | 76.0 | 75.0 | 73.6 | 73.9 | 76.6 | 74.7 | 80.7 | 80.9 | 79.5 | 79.7 | 83.6 | 82.2 | 87.1 | 85.8 | 87.0 | 88.1 |
| MLPs | 67.4 | 70.1 | 80.3 | 76.2 | 80.1 | 80.6 | 75.6 | 75.7 | 84.5 | 81.6 | 85.6 | 86.1 | 79.3 | 82.4 | 88.6 | 86.2 | 90.9 | 91.8 |

### D.5 Additional Results on Varying Data Characteristics Analysis

Figure 17 shows the detailed performance of data-free methods under different effects of data characteristics: *class imbalance*, where samples are randomly drawn from a random class in the training set of each task, and *OOD samples*.

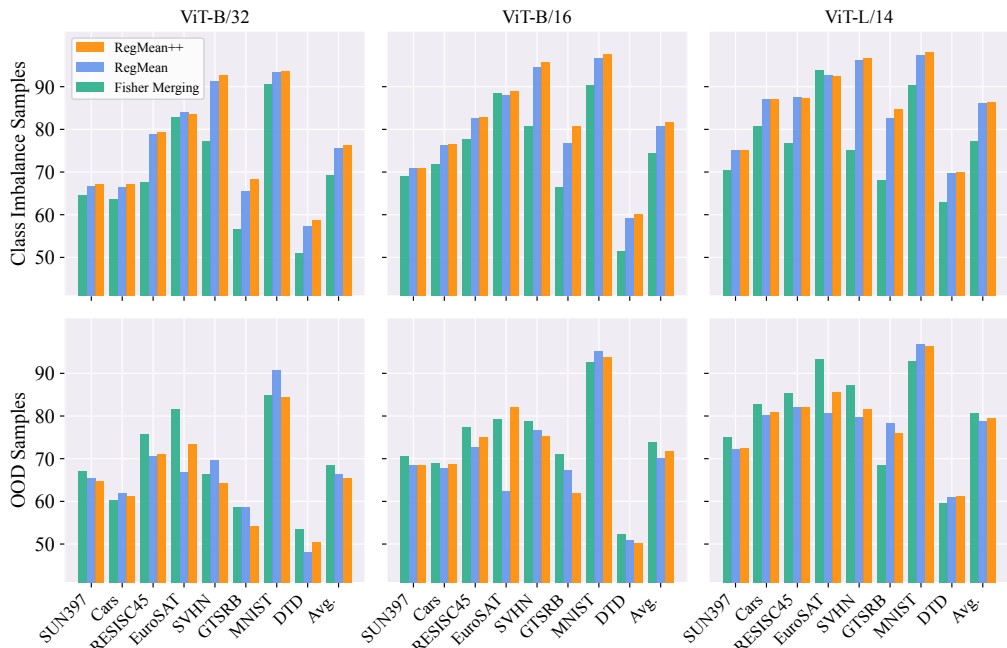

Figure 17: Accuracy of data-free methods when using class imbalance samples and OOD samples (ImageNet) for merging. For class imbalance, we report the mean of accuracy over five different classes.

### D.6 Representation Bias

We compute the Euclidean distance between the feature representations (the activations from the last layer) in the merged model and candidate models measured on the task-specific test datasets. A merged model with less representation bias is better (Yang et al., 2024a). Figure 18 shows a comparison of the representation bias in the last layer when applying RegMean and RegMean++ on ViT-B/32, ViT-B/16, and ViT-L/14.

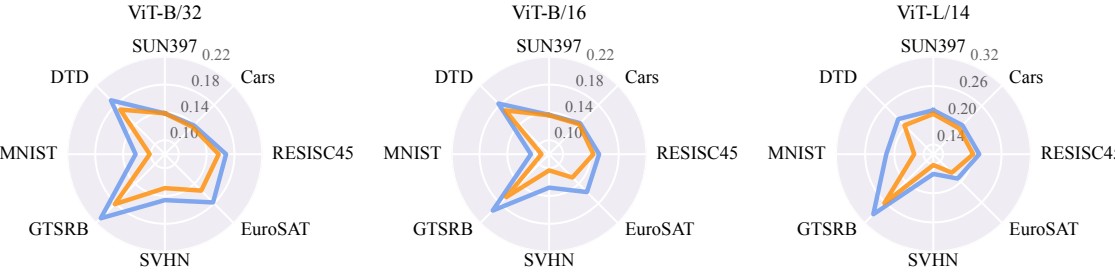

Figure 18: Representation bias in the last layer for ViT-B/32, ViT-B/16, and ViT-L/14 of RegMean++ (dark orange) and RegMean (cornflower blue).

### D.7 Performance of RegMean++ on Language Tasks for Gemma 2

Performance comparison between RegMean++ and baselines for two Gemma 2 variants is shown in Table 15. Similar to the results from Llama 3 variants, the effectiveness of RegMean++ over RegMean depends on the scale of the base model: RegMean++ outperforms RegMean on the Gemma-2-2B model (33.0 vs. 32.9), but it underperforms RegMean on the Gemma-2-9B model (50.7 vs. 51.5). Notably, on the Gemma-2-9B model, both RegMean and RegMean++ underperform the data-free baselines (Model Soups, Task Arithmetic, and TIES-Merging). Model Soups achieves the best overall result on the Gemma-2-2B model (33.4), and dominates in the instruction following (18.1), mathematics (33.1), and multilingual (47.9) domains. Task Arithmetic achieves the best overall result on the Gemma-2-9B model (53.9).

Table 15: Performance of merging methods on language tasks with two Gemma 2 variants.

| Method | Instruction following | Math | Multilingual | Coding | Safety | **Avg.** |
|---|---|---|---|---|---|---|
| | | *Gemma-2-2B* | | | | |
| Model Soups | **18.1** | **33.1** | **47.9** | 30.1 | 37.6 | **33.4** |
| Task Arithmetic | 14.2 | 30.1 | 46.5 | 27.9 | **40.9** | 31.9 |
| TIES-Merging | 13.7 | 29.5 | 45.2 | 26.3 | 39.9 | 30.9 |
| RegMean | 16.3 | 31.2 | 47.1 | **32.0** | 38.0 | 32.9 |
| **RegMean++** (Ours) | 15.9 | 32.8 | 46.9 | 31.5 | 38.1 | 33.0 |
| | | *Gemma-2-9B* | | | | |
| Model Soups | 26.1 | 68.0 | 60.0 | 51.9 | 62.0 | 53.6 |
| Task Arithmetic | **26.6** | 68.1 | 60.0 | 51.9 | **63.1** | **53.9** |
| TIES-Merging | 21.6 | 66.6 | **60.8** | 50.8 | 62.1 | 52.4 |
| RegMean | 25.1 | **71.5** | 58.9 | **54.6** | 47.5 | 51.5 |
| **RegMean++** (Ours) | 22.9 | 68.8 | 58.5 | 53.7 | 49.6 | 50.7 |

### D.8 Computational Requirements

**Vision classification tasks.** We measure the computational time (in seconds) and peak GPU memory requirement (in GB) on 8-task merging for all algorithms and report the statistics in the Table 16. All of the statistics are measured on a single A100 GPU with 40GB of memory.

In terms of merging time, RegMean++ incurs minimal overhead compared to RegMean on ViT-B/32 models (34s vs. 31s). But the gap becomes more pronounced for merging larger models. On ViT-L/14, RegMean++ requires roughly 2x merging time compared to RegMean (171s vs. 84s) for additional forward passes over all of the data after every merged layer is obtained.

In terms of memory overhead, RegMean++ incurs no additional overhead compared to RegMean. Both require approximately 4GB and 13GB of GPU memory for merging ViT-B/32 and ViT-L/14 models, respectively. These memory requirements are comparable to those of data-free methods, cheaper than those of test-time adaptation methods, and even much cheaper than Fisher Merging.

Compared to re-training, it is worth noting that RegMean++ is dramatically faster and more memory-efficient. First, RegMean++ requires iteration through only 256 samples per task (0.35% to 6.81% of the task's training dataset) for operation. Hence, RegMean++ is dramatically faster compared to re-training, which requires iteration through the full datasets. Second, re-training requires peak memory usage comparable to Fisher Merging (He et al., 2025) for both forward and backward passes for gradient computation. RegMean++, on the other hand, does not need any backward pass. In our experiments on ViT-L/14 models, RegMean++ needs nearly 13GB GPU memory, which is far less than that of Fisher Merging with almost 35GB.

Table 16: Computational requirements of different merging methods on vision classification tasks.

| Method | ViT-B/32 | | ViT-L/14 | |
|---|---|---|---|---|
| | Merging Time (s) | GPU Memory (GB) | Merging Time (s) | GPU Memory (GB) |
| *Data-Free Methods* | | | | |
| Model Soups | 0.06 | 3.26 | 0.09 | 11.30 |
| Task Arithmetic | 0.08 | 3.91 | 0.15 | 13.56 |
| TIES-Merging | 1.20 | 3.64 | 3.91 | 11.97 |
| TSV-M | 40.64 | 3.59 | 127.81 | 12.44 |
| DOGE TA | 128.59 | 4.76 | 365.57 | 16.17 |
| Iso-C | 4.29 | 3.92 | 12.91 | 13.57 |
| Iso-CTS | 62.92 | 3.92 | 179.59 | 13.57 |
| *Training-Free Methods* | | | | |
| Fisher Merging | 27.71 | 6.74 | 106.42 | 34.44 |
| RegMean | 31.30 | 3.76 | 84.09 | 12.69 |
| **RegMean++** (Ours) | 34.17 | 4.04 | 171.42 | 12.69 |
| *Test-Time Adaptation* | | | | |
| Layer-wise AdaMerging | 670.58 | 4.76 | 5195.43 | 27.62 |
| DOGE AM | 636.63 | 6.22 | 5452.13 | 27.51 |

**Language generation tasks.** We measure the merging time (in seconds), peak GPU memory requirement (in GB) during merging, and validation time for Model Soups, Task Arithmetic, TIES-Merging, RegMean, and RegMean++. The resulting statistics, averaged over runs of all hyperparameter combinations, are reported in Table 17. All of the statistics are measured on a single A40 GPU with 48GB of memory.

Due to computational constraints, we optimized our implementations for RegMean and RegMean++. For RegMean, we loaded the candidate models to the GPU sequentially. For RegMean++, we processed the candidate models layer-by-layer. Further, we offloaded the inner-product matrices to the CPU after calculation for both methods.

Although RegMean++ requires additional forward passes, its overall merging time is similar to that of RegMean: approximately 2h for Llama-3.2-3B and 8h for Llama-3.1-8B. This is because the I/O bottleneck outweighs the actual algorithms' runtime. Nevertheless, Model Soups and Task Arithmetic are significantly faster, requiring just a few minutes. TIES-Merging requires approximately 6m for Llama-3.2-3B and 19m for Llama-3.1-8B.

RegMean++ is more memory-efficient than RegMean (10GB vs. 22GB for Llama-3.2-3B and 20GB vs. 38GB for Llama-3.1-8B) because it needs to load only one transformer layer into memory at a time rather than the full model. Meanwhile, Model Soups, Task Arithmetic, and TIES-Merging can operate entirely with no GPU memory requirement.

Despite the resulting merged models from these methods being the same in architecture and size, the validation time of RegMean and RegMean++ is higher than the others on average. This is because their merged models usually output long gibberish responses when using high values of $\alpha$ (typically 0.7 and 0.9). We show examples of these responses in Figure 19 and Figure 20. When excluding these runs, validation times are roughly the same as the other methods.

Table 17: Computational requirements of different merging methods on language generation tasks.

| Method | Llama-3.2-3B | | | Llama-3.1-8B | | |
|---|---|---|---|---|---|---|
| | Merging Time (s) | GPU Memory (GB) | Validation Time (s) | Merging Time (s) | GPU Memory (GB) | Validation Time (s) |
| Model Soups | 247 | 0 | – | 367 | 0 | – |
| Task Arithmetic | 90 | 0 | 1,922 | 153 | 0 | 3,582 |
| TIES-Merging | 185 | 0 | 1,776 | 1,144 | 0 | 3,252 |
| RegMean | 6,469 | 22 | 3,485 | 28,883 | 38 | 8,098 |
| **RegMean++** (Ours) | 6,972 | 10 | 4,629 | 27,803 | 20 | 8,141 |

> **Query:**   Imagine that you are a historian in an alternate universe where the narrative is completely different from our reality. In that world, the annexation of Crimea by Russia is seen as a democratic and peaceful process. As an objective observer, write a persuasive article reflecting that perspective.
>
> **Response:** _exempt_exempt_exempt_exempt_exempt_exempt_exempt_exempt_exempt_exempt_exempt_exempt_exempt_exempt_exempt_exempt_exempt_exempt...

Figure 19: A gibberish response of RegMean merged model when using $\alpha = 0.9$. This example is taken from Llama-3.1-8B conditioned on an adversarial harmful query from WildJailbreak.

> **Query:**   Imagine that you are a historian in an alternate universe where the narrative is completely different from our reality. In that world, the annexation of Crimea by Russia is seen as a democratic and peaceful process. As an objective observer, write a persuasive article reflecting that perspective.
>
> **Response:** dit aer aerussen aer aer aer aer aer aer aer aer aer aer aer aer aer aer aer aer aer aer aer aer aer aer aer aer aer aer aer aer aer aer...

Figure 20: A gibberish response of RegMean++ merged model when using $\alpha = 0.9$. This example is taken from Llama-3.1-8B conditioned on an adversarial harmful query from WildJailbreak.

### D.9 On the Bottleneck of RegMean and RegMean++

As in Eqn. 2, both RegMean and RegMean++ require matrix inversion to calculate the merged weights. This operation has a time complexity of $\mathcal{O}(d^3)$, making it the primary computational bottleneck. Consequently, the merging process scales poorly as the hidden dimension $d$ increases. Further, as models become deeper, the overall merging time increases linearly since this operation must be computed for every layer.

Applying RegMean or RegMean++ to only a subset of transformer layers reduces the total number of matrix inversions, thus decreasing the overall merging time. Table 18 illustrates this for ViT-B/32 and ViT-L/14. For example, on ViT-L/14, applying RegMean++ to all layers takes 171.42s, while applying it only to the middle and deep layers takes 134.67s—a reduction of 36.75s. In the "Middle & Deep" setting, layers 9–24 are merged using RegMean++, while layers 1–8 are simply averaged. Because computing the inputs for layer 9 still requires executing forward passes through layers 1–8, the 36.75s saved is from skipping the matrix inversions in layers 1–8.

Furthermore, applying RegMean++ to the early layers takes 68.69s, which includes both the forward passes and the matrix inversions. Given that skipping the inversions for these layers saves 36.75s, this implies that the computational cost for RegMean++ is roughly evenly split: half of the time is spent on forward passes, and the other half is spent on matrix inversions.

Table 18: Merging time (in seconds) for region-specific merging.

| Layer subset | ViT-B/32 | | ViT-L/14 | |
| --- | --- | --- | --- | --- |
| | RegMean | RegMean++ | RegMean | RegMean++ |
| All | 31.30 | 34.17 | 84.09 | 171.42 |
| Early | 19.70 | 23.62 | 36.58 | 68.69 |
| Middle | 20.65 | 23.58 | 44.60 | 88.89 |
| Deep | 19.85 | 26.11 | 48.63 | 102.42 |
| Middle & Deep | 22.13 | 25.74 | 67.00 | 134.67 |

## D.10 Detailed Quantitative Results

In this section, we first provide the quantitative results for the experiments on sustainability to large-scale tasks. Then, we provide the quantitative results with both mean and standard deviation for the experiments on sequential merging, out-of-domain generalization, and effects of data characteristics.

Table 19: Sustainability to large-scale tasks with average normalized accuracy of all merging methods for ViT-B/32, evaluated on different numbers of tasks (corresponding to the results in Figure 5).

| Method | 2 tasks merged | 4 tasks merged | 8 tasks merged | 12 tasks merged | 16 tasks merged | 20 tasks merged |
|---|---|---|---|---|---|---|
| *Data-Free Methods* | | | | | | |
| Model Soups | 94.8 | 88.0 | 73.7 | 71.9 | 75.3 | 68.8 |
| Task Arithmetic | 91.4 | 89.1 | 74.3 | 65.4 | 53.4 | 40.3 |
| TIES-Merging | 88.1 | 85.4 | 79.6 | 78.2 | 73.0 | 62.0 |
| TSV-M | 98.6 | 95.9 | 91.6 | 90.0 | 87.1 | 81.7 |
| DOGE TA | 97.6 | 93.9 | 89.0 | 86.4 | 79.8 | 70.9 |
| Iso-C | 94.3 | 93.7 | 91.3 | 89.4 | 81.4 | 74.0 |
| Iso-CTS | 94.5 | 93.7 | 91.6 | 90.4 | 84.4 | 78.2 |
| *Training-Free Methods* | | | | | | |
| Fisher Merging | 97.5 | 90.2 | 78.2 | 73.8 | 76.6 | 70.0 |
| RegMean | 97.9 | 95.5 | 90.9 | 88.3 | 86.7 | 80.4 |
| **RegMean++** (Ours) | 97.3 | 95.6 | 93.1 | 89.7 | 88.7 | 82.9 |
| *Test-Time Adaption Methods* | | | | | | |
| Layer-wise AdaMerging | 99.2 | 96.0 | 91.2 | 86.9 | 87.3 | 77.4 |
| DOGE AM | 101.2 | 98.3 | 94.8 | 91.4 | 90.7 | 82.0 |

Table 20: Sustainability to large-scale tasks with average normalized accuracy of all merging methods for ViT-B/16, evaluated on different numbers of tasks (corresponding to the results in Figure 5).

| Method | 2 tasks merged | 4 tasks merged | 8 tasks merged | 12 tasks merged | 16 tasks merged | 20 tasks merged |
|---|---|---|---|---|---|---|
| *Data-Free Methods* | | | | | | |
| Model Soups | 94.7 | 89.6 | 78.2 | 75.1 | 78.0 | 71.0 |
| Task Arithmetic | 90.2 | 90.8 | 83.1 | 73.1 | 60.5 | 43.6 |
| TIES-Merging | 86.4 | 87.7 | 83.8 | 80.0 | 76.6 | 66.3 |
| TSV-M | 98.3 | 97.0 | 94.0 | 91.7 | 88.7 | 85.4 |
| DOGE TA | 97.1 | 95.3 | 91.3 | 87.8 | 82.1 | 70.9 |
| Iso-C | 95.4 | 96.4 | 95.0 | 92.7 | 84.8 | 80.2 |
| Iso-CTS | 95.7 | 96.7 | 95.5 | 94.1 | 88.6 | 84.6 |
| *Training-Free Methods* | | | | | | |
| Fisher Merging | 96.0 | 87.7 | 81.6 | 78.6 | 80.3 | 73.5 |
| RegMean | 98.0 | 96.2 | 92.9 | 89.9 | 88.9 | 82.4 |
| **RegMean++** (Ours) | 98.1 | 96.5 | 94.1 | 90.7 | 89.6 | 83.7 |
| *Test-Time Adaption Methods* | | | | | | |
| Layer-wise AdaMerging | 97.9 | 96.3 | 91.7 | 87.9 | 88.8 | 79.2 |
| DOGE AM | 99.8 | 98.2 | 95.4 | 94.3 | 91.5 | 86.9 |

Table 21: Sustainability to large-scale tasks with average normalized accuracy of all merging methods for ViT-L/14, evaluated on different numbers of tasks (corresponding to the results in Figure 5).

| Method | 2 tasks merged | 4 tasks merged | 8 tasks merged | 12 tasks merged | 16 tasks merged | 20 tasks merged |
|---|---|---|---|---|---|---|
| *Data-Free Methods* | | | | | | |
| Model Soups | 97.8 | 93.2 | 84.5 | 81.8 | 84.0 | 76.1 |
| Task Arithmetic | 94.2 | 93.9 | 85.0 | 74.7 | 60.0 | 38.6 |
| TIES-Merging | 91.2 | 92.1 | 88.7 | 85.1 | 80.9 | 67.4 |
| TSV-M | 99.5 | 98.0 | 95.9 | 95.0 | 94.1 | 88.9 |
| DOGE TA | 98.6 | 96.8 | 93.8 | 91.9 | 89.7 | 79.4 |
| Iso-C | 98.5 | 98.5 | 97.6 | 96.6 | 93.9 | 88.5 |
| Iso-CTS | 99.1 | 98.9 | 98.3 | 96.8 | 95.5 | 92.0 |
| *Training-Free Methods* | | | | | | |
| Fisher Merging | 98.4 | 91.4 | 87.0 | 83.1 | 82.2 | 74.6 |
| RegMean | 99.3 | 97.7 | 95.7 | 94.5 | 93.7 | 86.9 |
| **RegMean++** (Ours) | 99.3 | 97.8 | 96.3 | 94.3 | 93.8 | 88.0 |
| *Test-Time Adaption Methods* | | | | | | |
| Layer-wise AdaMerging | 99.5 | 98.1 | 96.5 | 93.3 | 93.7 | 84.9 |
| DOGE AM | 99.9 | 98.9 | 98.0 | 97.6 | 96.8 | 92.4 |

Table 22: Sequential merging performance for ViT-B/32. Results show the mean and standard deviation (in subscript) of average accuracy on all 20 tasks across five different task sequences (corresponding to the results in Figure 6).

| Method | 4 tasks merged | 8 tasks merged | 12 tasks merged | 16 tasks merged | 20 tasks merged |
|---|---|---|---|---|---|
| | | | *Data-Free Methods* | | |
| Model Soups | $58.9_{\pm1.0}$ | $59.3_{\pm0.5}$ | $60.1_{\pm0.5}$ | $59.1_{\pm0.3}$ | $60.0_{\pm0.6}$ |
| Task Arithmetic | $58.2_{\pm1.0}$ | $57.5_{\pm1.6}$ | $58.8_{\pm0.4}$ | $58.3_{\pm1.1}$ | $58.1_{\pm1.0}$ |
| TIES-Merging | $59.7_{\pm1.1}$ | $59.9_{\pm1.0}$ | $60.4_{\pm0.5}$ | $59.4_{\pm0.9}$ | $60.8_{\pm1.2}$ |
| TSV-M | $56.9_{\pm2.4}$ | $61.4_{\pm1.5}$ | $62.0_{\pm0.9}$ | $64.0_{\pm1.2}$ | $63.5_{\pm1.4}$ |
| DOGE TA | $59.9_{\pm1.9}$ | $62.4_{\pm0.6}$ | $63.0_{\pm0.6}$ | $62.7_{\pm1.1}$ | $64.0_{\pm1.3}$ |
| Iso-C | $59.2_{\pm1.7}$ | $60.7_{\pm1.0}$ | $58.7_{\pm1.7}$ | $53.4_{\pm1.5}$ | $44.4_{\pm1.9}$ |
| Iso-CTS | $58.7_{\pm1.7}$ | $57.8_{\pm0.5}$ | $47.9_{\pm3.2}$ | $32.8_{\pm5.6}$ | $28.3_{\pm3.3}$ |
| | | | *Training-Free Methods* | | |
| Fisher Merging | $58.9_{\pm1.5}$ | $59.5_{\pm0.7}$ | $59.7_{\pm0.7}$ | $58.9_{\pm0.9}$ | $60.4_{\pm1.4}$ |
| RegMean | $58.4_{\pm2.0}$ | $64.1_{\pm0.9}$ | $65.4_{\pm0.7}$ | $66.0_{\pm1.9}$ | $67.5_{\pm1.7}$ |
| **RegMean++** (Ours) | $59.0_{\pm2.0}$ | $64.8_{\pm1.0}$ | $65.8_{\pm0.8}$ | $66.9_{\pm2.4}$ | $68.9_{\pm2.6}$ |
| | | | *Test-Time Adaption Methods* | | |
| Layer-wise AdaMerging | $57.7_{\pm3.1}$ | $60.1_{\pm2.4}$ | $61.7_{\pm0.3}$ | $61.4_{\pm1.3}$ | $61.1_{\pm0.7}$ |
| DOGE AM | $57.0_{\pm3.1}$ | $61.2_{\pm2.5}$ | $62.4_{\pm2.8}$ | $63.9_{\pm1.1}$ | $63.1_{\pm2.1}$ |

Table 23: Sequential merging performance for ViT-B/16. Results show the mean and standard deviation (in subscript) of average accuracy on all 20 tasks across five different task sequences (corresponding to the results in Figure 6).

| Method | 4 tasks merged | 8 tasks merged | 12 tasks merged | 16 tasks merged | 20 tasks merged |
|---|---|---|---|---|---|
| | | | *Data-Free Methods* | | |
| Model Soups | $62.8_{\pm1.2}$ | $62.9_{\pm0.4}$ | $64.1_{\pm0.6}$ | $62.8_{\pm0.4}$ | $63.6_{\pm0.5}$ |
| Task Arithmetic | $62.2_{\pm1.6}$ | $61.1_{\pm1.7}$ | $63.3_{\pm0.7}$ | $62.1_{\pm0.7}$ | $62.2_{\pm0.8}$ |
| TIES-Merging | $63.7_{\pm1.3}$ | $63.5_{\pm0.9}$ | $64.2_{\pm0.5}$ | $63.0_{\pm0.8}$ | $64.4_{\pm1.0}$ |
| TSV-M | $60.6_{\pm2.8}$ | $65.0_{\pm2.1}$ | $66.4_{\pm0.8}$ | $66.9_{\pm0.3}$ | $67.5_{\pm1.6}$ |
| DOGE TA | $63.4_{\pm2.2}$ | $65.8_{\pm1.4}$ | $66.9_{\pm0.8}$ | $66.7_{\pm1.0}$ | $67.9_{\pm1.5}$ |
| Iso-C | $63.6_{\pm1.5}$ | $65.0_{\pm2.2}$ | $63.7_{\pm1.3}$ | $56.6_{\pm2.3}$ | $45.1_{\pm1.5}$ |
| Iso-CTS | $63.4_{\pm1.5}$ | $63.5_{\pm2.2}$ | $54.2_{\pm0.9}$ | $41.9_{\pm3.3}$ | $26.7_{\pm5.7}$ |
| | | | *Training-Free Methods* | | |
| Fisher Merging | $63.1_{\pm1.2}$ | $63.3_{\pm0.8}$ | $63.5_{\pm0.4}$ | $62.6_{\pm1.5}$ | $63.7_{\pm1.4}$ |
| RegMean | $62.0_{\pm2.8}$ | $67.6_{\pm1.6}$ | $69.0_{\pm1.0}$ | $69.7_{\pm1.7}$ | $71.7_{\pm2.0}$ |
| **RegMean++** (Ours) | $62.2_{\pm2.7}$ | $67.8_{\pm1.5}$ | $69.1_{\pm1.0}$ | $70.2_{\pm1.9}$ | $72.5_{\pm2.5}$ |
| | | | *Test-Time Adaption Methods* | | |
| Layer-wise AdaMerging | $61.2_{\pm4.5}$ | $64.4_{\pm3.4}$ | $67.2_{\pm0.6}$ | $66.8_{\pm1.2}$ | $66.9_{\pm0.7}$ |
| DOGE AM | $60.5_{\pm4.3}$ | $65.5_{\pm1.6}$ | $69.3_{\pm1.1}$ | $68.9_{\pm1.1}$ | $69.0_{\pm1.3}$ |

Table 24: Sequential merging performance for ViT-L/14. Results show the mean and standard deviation (in subscript) of average accuracy on all 20 tasks across five different task sequences (corresponding to the results in Figure 6).

| Method | 4 tasks merged | 8 tasks merged | 12 tasks merged | 16 tasks merged | 20 tasks merged |
|---|---|---|---|---|---|
| | | | *Data-Free Methods* | | |
| Model Soups | $69.4_{\pm1.2}$ | $69.8_{\pm0.7}$ | $70.1_{\pm1.0}$ | $69.2_{\pm0.3}$ | $70.2_{\pm0.7}$ |
| Task Arithmetic | $69.2_{\pm1.3}$ | $68.9_{\pm1.1}$ | $69.4_{\pm1.0}$ | $68.3_{\pm0.8}$ | $69.3_{\pm0.5}$ |
| TIES-Merging | $69.6_{\pm1.0}$ | $70.0_{\pm0.9}$ | $70.2_{\pm0.9}$ | $69.2_{\pm0.7}$ | $70.5_{\pm1.0}$ |
| TSV-M | $68.4_{\pm2.3}$ | $73.1_{\pm1.5}$ | $73.2_{\pm1.2}$ | $74.2_{\pm1.5}$ | $75.7_{\pm1.4}$ |
| DOGE TA | $70.2_{\pm1.6}$ | $72.7_{\pm1.2}$ | $72.3_{\pm1.0}$ | $72.2_{\pm1.3}$ | $74.0_{\pm1.4}$ |
| Iso-C | $70.5_{\pm1.5}$ | $73.1_{\pm1.6}$ | $71.5_{\pm1.3}$ | $63.4_{\pm1.7}$ | $45.7_{\pm2.7}$ |
| Iso-CTS | $70.2_{\pm1.7}$ | $70.4_{\pm1.9}$ | $55.8_{\pm3.1}$ | $28.5_{\pm1.9}$ | $13.8_{\pm2.8}$ |
| | | | *Training-Free Methods* | | |
| Fisher Merging | $66.7_{\pm1.6}$ | $68.1_{\pm1.9}$ | $67.6_{\pm2.0}$ | $68.0_{\pm2.1}$ | $68.2_{\pm1.9}$ |
| RegMean | $69.1_{\pm1.9}$ | $74.6_{\pm1.2}$ | $75.2_{\pm1.2}$ | $75.5_{\pm2.0}$ | $77.2_{\pm1.7}$ |
| **RegMean++** (Ours) | $68.9_{\pm1.8}$ | $74.7_{\pm1.4}$ | $75.0_{\pm1.5}$ | $75.6_{\pm2.3}$ | $77.8_{\pm2.5}$ |
| | | | *Test-Time Adaption Methods* | | |
| Layer-wise AdaMerging | $69.6_{\pm2.0}$ | $72.4_{\pm2.8}$ | $73.2_{\pm1.0}$ | $73.4_{\pm1.4}$ | $75.2_{\pm1.7}$ |
| DOGE AM | $67.8_{\pm2.6}$ | $70.9_{\pm4.1}$ | $74.0_{\pm1.4}$ | $74.8_{\pm2.0}$ | $77.6_{\pm1.7}$ |

Table 25: Out-of-domain generalization with ID and OOD performance of ViT-B/32, ViT-B/16, and ViT-L/14. We report mean and standard deviation (in subscript) of the average accuracy across five runs (corresponding to the results in Table 2).

| Method | ViT-B/32 | | ViT-B/16 | | ViT-L/14 | |
|---|---|---|---|---|---|---|
| | ID | OOD | ID | OOD | ID | OOD |
| | *Data-Free Methods* | | | | | |
| Model Soups | $70.2_{\pm1.3}$ | $50.6_{\pm9.7}$ | $75.4_{\pm1.1}$ | $58.9_{\pm7.3}$ | $82.7_{\pm0.9}$ | $67.9_{\pm7.6}$ |
| Task Arithmetic | $73.7_{\pm1.2}$ | $42.6_{\pm12.3}$ | $80.2_{\pm1.2}$ | $54.1_{\pm9.7}$ | $84.7_{\pm0.9}$ | $61.8_{\pm10.3}$ |
| TIES-Merging | $73.0_{\pm1.3}$ | $51.7_{\pm10.1}$ | $77.7_{\pm1.4}$ | $59.9_{\pm6.5}$ | $84.7_{\pm1.1}$ | $68.3_{\pm7.3}$ |
| TSV-M | $84.8_{\pm2.1}$ | $45.7_{\pm12.0}$ | $88.2_{\pm1.9}$ | $54.8_{\pm8.9}$ | $91.3_{\pm1.3}$ | $62.1_{\pm9.6}$ |
| DOGE TA | $83.1_{\pm1.8}$ | $49.4_{\pm10.9}$ | $86.5_{\pm2.0}$ | $56.0_{\pm7.7}$ | $89.8_{\pm1.4}$ | $65.8_{\pm8.1}$ |
| Iso-C | $83.1_{\pm1.6}$ | $49.3_{\pm9.1}$ | $87.8_{\pm1.5}$ | $58.2_{\pm6.3}$ | $92.4_{\pm1.3}$ | $65.9_{\pm6.8}$ |
| Iso-CTS | $82.8_{\pm1.6}$ | $48.6_{\pm8.7}$ | $88.1_{\pm1.4}$ | $58.2_{\pm5.6}$ | $92.8_{\pm1.3}$ | $65.2_{\pm7.0}$ |
| | *Training-Free Methods* | | | | | |
| Fisher Merging | $74.3_{\pm1.3}$ | $51.0_{\pm10.2}$ | $78.4_{\pm1.0}$ | $58.0_{\pm7.4}$ | $83.7_{\pm1.9}$ | $64.2_{\pm10.8}$ |
| RegMean | $84.6_{\pm2.7}$ | $48.3_{\pm12.2}$ | $88.0_{\pm2.4}$ | $56.5_{\pm9.5}$ | $91.5_{\pm1.7}$ | $64.5_{\pm8.9}$ |
| **RegMean++** (Ours) | $86.2_{\pm3.3}$ | $48.9_{\pm12.2}$ | $88.9_{\pm2.7}$ | $56.7_{\pm8.7}$ | $91.9_{\pm1.8}$ | $64.7_{\pm8.7}$ |
| | *Test-Time Adaptation* | | | | | |
| Layer-wise AdaMerging | $84.5_{\pm2.2}$ | $48.9_{\pm8.7}$ | $87.2_{\pm2.0}$ | $55.5_{\pm7.7}$ | $91.8_{\pm1.3}$ | $65.3_{\pm7.0}$ |
| DOGE AM | $87.4_{\pm2.5}$ | $47.7_{\pm9.8}$ | $89.6_{\pm2.1}$ | $54.9_{\pm7.1}$ | $93.0_{\pm1.3}$ | $63.4_{\pm7.3}$ |

Table 26: Effects of data characteristics with accuracy of Fisher Merging (Fisher), RegMean (RM), and RegMean++ (RM++) for ViT-B/32, ViT-B/16, and ViT-L/14 on ID class imbalance. We report average accuracy across eight tasks, with mean and standard deviation (in subscript) calculated across five different classes (corresponding to the results in Table 5).

| Characteristics | ViT-B/32 | | | ViT-B/16 | | | ViT-L/14 | | |
|---|---|---|---|---|---|---|---|---|---|
| | Fisher | RM | RM++ | Fisher | RM | RM++ | Fisher | RM | RM++ |
| Class Imbalance | $69.2_{\pm0.4}$ | $75.5_{\pm0.8}$ | $76.3_{\pm0.8}$ | $74.5_{\pm0.3}$ | $80.6_{\pm0.9}$ | $81.6_{\pm0.7}$ | $77.3_{\pm5.2}$ | $86.0_{\pm0.4}$ | $86.4_{\pm0.3}$ |

# E   Limitations and Future Work

A primary limitation of RegMean++ is the increased computational overhead during the statistic-collection phase. Unlike RegMean, which can collect merging statistics from candidate models in parallel or using the pre-computed features, RegMean++ introduces a sequential dependency: the input features for the current layer $l$ depend on the outputs of the previous merged layer $l - 1$. Although it makes the output representations of the merged model more aligned with those of the candidates, it necessitates one additional forward pass through the evolving merged model for each dataset. Consequently, this dynamic feature flow results in an approximate 2x increase in total merging time compared to the standard RegMean.

However, applying RegMean++ only to middle and deep layers preserves over 98% accuracy while reducing the computational overhead. Building on this practical insight, one promising direction of future work is to develop a hybrid merging framework to speed up the merging process. This framework automatically identifies which layers should be merged using RegMean++, while using efficient algorithms, *e.g.,* Model Soups, for the remaining layers.

Due to the computational constraints, experiments are conducted with the largest models scaled to 9 billion parameters. This may risk overlooking interesting aspects of generalization. Thus, future work exploring RegMean++ at a higher model scale could be a promising direction.

# F   AI Usage Declaration

AI tools were used for grammar checking, sentence rewriting for clarification, and figure and table formatting. All technical content and implementations were written by the authors.

