# OpenReview forum: "RegMean++: Enhancing Effectiveness and Generalization of Regression Mean for Model Merging"
_TMLR — Accepted by TMLR_

### Review · Reviewer_NQcZ · 2026-03-08

**Summary Of Contributions:**

- This paper proposes RegMean++, an improvement of RegMean for model merging. It changes how the layerwise regression statistics are collected: instead of using candidate-model activations as inputs for each layer’s closed-form regression, RegMean++ uses merge-model activations propagated through the already-merged lower layers, aiming to better reflect intra-layer and cross-layer dependencies during merging.

- The paper’s core empirical results are strongest on vision benchmarks (CLIP ViT variants on 8-task and 20-task settings), where RegMean++ consistently improves over RegMean in average accuracy and robustness to common corruptions—often by modest but repeatable margins. The evidence for OOD generalization (defined as holding out tasks from the same benchmark) shows small gains over RegMean, and RegMean++ is not always the top OOD method in their comparison.

**Audience:**

Yes

**Audience Explanation:**

model merging is an interesting topic to people in the community.

**Claims And Evidence:**

Yes

**Claims Explanation:**

- the claims about vision merging improvements and better alignment of merge-model representations (CKA) are generally supported by the presented tables/figures; broader claims about consistent gains across settings including language tasks are only partially supported given the reported regressions and limited baseline coverage in the language section.

**Requested Changes:**

- Language baseline coverage: Given MergeBench explicitly evaluates multiple merging methods and discusses scaling constraints, it would be helpful (and arguably necessary for strong claims) to compare RegMean++ against the same baseline set as vision, at least for a subset.

- Hyperparameter selection protocol clarity (especially language): The procedure for selecting $\alpha$ should state what data split is used for tuning to avoid test leakage.

- Uncertainty quantification / run-to-run variance: Several improvements are small (especially OOD), and without consistent error bars or statistical tests it is hard to know if differences are robust across seeds/task splits.

---

> ### Author Response · Authors · 2026-03-23
> **Official Comments by Authors (1/2)**
>
> We thank the Reviewer for your comments, effort, and time in reviewing our paper. Please find our responses to your concerns below. We have also updated our manuscript; the added contents are highlighted in blue.
>
> ---
>
> > Language baseline coverage: Given MergeBench explicitly evaluates multiple merging methods and discusses scaling constraints, it would be helpful (and arguably necessary for strong claims) to compare RegMean++ against the same baseline set as vision, at least for a subset.
>
> **A:** We have extended our language baselines by including Model Soups, Task Arithmetic, and TIES-Merging.
>
> **Performance comparison.** We report results in the two tables below. We find that TIES-Merging achieves the highest overall merging performance for both Llama-3.2-3B and Llama-3.1-8B. Model Soups and Task Arithmetic demonstrate comparable or even higher merging performance compared to RegMean and RegMean++. *We also conducted experiments on Gemma-2-2B and Gemma-2-9B. Please take a look at our General Response: Additional Results on Gemma 2 Models.*
>
> *Performance comparison on Llama-3.2-3B:*
>
> |Method|Instruction following|Math|Multilingual|Coding|Safety|Avg.|
> |:-|:-:|:-:|:-:|:-:|:-:|:-:|
> |Model Soups|8.7|36.2|47.8|37.1|36.5|33.3|
> |Task Arithmetic|19.8|37.1|**48.1**|40.1|**43.6**|37.7|
> |TIES-Merging|**22.0**|**42.8**|**48.1**|**40.5**|41.2|**38.9**|
> |RegMean|8.3|35.5|47.3|39.2|39.8|34.0|
> |**RegMean++** (Ours)|5.9|32.1|47.1|38.1|36.3|31.9|
>
> *Performance comparison on Llama-3.1-8B:*
>
> |Method|Instruction following|Math|Multilingual|Coding|Safety|Avg.|
> |:-|:-:|:-:|:-:|:-:|:-:|:-:|
> |Model Soups|13.9|**67.9**|54.4|50.2|56.2|48.5|
> |Task Arithmetic|8.7|62.2|54.8|47.5|53.5|45.3|
> |TIES-Merging|13.1|67.1|**54.9**|49.2|**59.5**|**48.8**|
> |RegMean|**25.9**|55.1|51.4|51.5|33.6|43.5|
> |**RegMean++** (Ours)|11.1|65.8|53.1|**52.3**|46.3|45.7|
>
> **Scaling constraints with computational requirements.** In the two tables below, we report the statistics of computational requirements for all language baselines, averaged over runs of all hyperparameter combinations. All of the statistics are measured on a single A40 GPU with 48GB of memory.
>
> Due to computational constraints, we optimized our implementations for RegMean and RegMean++. For RegMean, we loaded the candidate models to the GPU sequentially. For RegMean++, we processed the candidate models layer-by-layer. Further, we offloaded the inner-product matrices to the CPU after calculation for both methods.
>
> Although RegMean++ requires additional forward passes, its overall merging time is similar to that of RegMean: approximately 2h for Llama-3.2-3B and 8h for Llama-3.1-8B. This is because the I/O bottleneck outweighs the actual algorithms' runtime. Nevertheless, Model Soups and Task Arithmetic are significantly faster, requiring just a few minutes. TIES-Merging requires approximately 6m for Llama-3.2-3B and 19m for Llama-3.1-8B.
>
> RegMean++ is more memory-efficient than RegMean (10GB vs. 22GB for Llama-3.2-3B and 20GB vs. 38GB for Llama-3.1-8B) because it needs to load only one transformer layer into memory at a time rather than the full model. Meanwhile,  Model Soups, Task Arithmetic, and TIES-Merging can operate entirely with no GPU memory requirement.
>
> Despite the resulting merged models from these methods being the same in architecture and size, the validation time of RegMean and RegMean++ is higher than the others on average. This is because their merged models usually output long gibberish responses when using high values of $\alpha$ (typically 0.7 and 0.9). When excluding these runs, validation times are roughly the same as the other methods.
>
> *Computational requirements on Llama-3.2-3B:*
>
> |Method|Merging Time (s)|GPU Memory (GB)|Validation Time (s)|
> |:---|:---:|:---:|:---:|
> |Model Soups|247|0|-|
> |Task Arithmetic|90|0|1,922|
> |TIES-Merging|185|0|1,776|
> |RegMean|6,469|22|3,485|
> |**RegMean++** (Ours)|6,972|10|4,629|
>
> *Computational requirements on Llama-3.1-8B:*
>
> |Method|Merging Time (s)|GPU Memory (GB)|Validation Time (s)|
> |:---|:---:|:---:|:---:|
> |Model Soups|367|0|-|
> |Task Arithmetic|153|0|3,582|
> |TIES-Merging|1,144|0|3,252|
> |RegMean|28,883|38|8,098|
> |**RegMean++** (Ours)|27,803|20|8,141|

---

> > ### Author Response · Authors · 2026-03-23
> > **Official Comments by Authors (2/2)**
> >
> > > Hyperparameter selection protocol clarity (especially language): The procedure for selecting $\alpha$ should state what data split is used for tuning to avoid test leakage.
> >
> > **A:** **For vision tasks.** We select $\alpha$ based on the validation sets, which are the held-out 10\% of the training samples (see Appendix C.3).
> >
> > **For language tasks.** In our initial manuscript, we did not search $\alpha$ on dedicated validation sets; instead, we evaluated all values of $\alpha$ and reported the best result for each method. In the revised manuscript, we follow MergeBench [1] to tune $\alpha$ on surrogate tasks (see Appendix B.1). However, based on the codebase provided by MergeBench, we reuse IFEval as the validation benchmark for the instruction-following domain. We acknowledge test leakage on this domain.
> >
> > > Uncertainty quantification / run-to-run variance: Several improvements are small (especially OOD), and without consistent error bars or statistical tests it is hard to know if differences are robust across seeds/task splits.
> >
> > **A:** To further support our claims, we have added standard deviation for the experiments on sequential merging, out-of-domain generalization, and effects of data characteristics in Appendix D.10.
> >
> > [1] He, Yifei, et al. "MergeBench: A Benchmark for Merging Domain-Specialized LLMs." The Thirty-ninth Annual Conference on Neural Information Processing Systems Datasets and Benchmarks Track.
> >
> > ---
> >
> > We hope these responses address your concerns. Please take a look at our revised manuscript. If any concerns remain, we are happy to discuss further.

---

### Review · Reviewer_k673 · 2026-03-10

**Summary Of Contributions:**

The authors introduce RegMean++, a method that improves upon the existing Regression Mean (RegMean) approach for model merging. While standard RegMean solves a linear regression problem layer-by-layer independently, RegMean++ explicitly models intra-layer and cross-layer dependencies by computing inner-product matrices using activations propagated through the currently merged layers rather than the original candidate layers. The authors demonstrate that this alignment strategy yields consistent improvements in accuracy across various vision (ViT variants) and language tasks (Llama 3 variants).

**Strengths:**

1. The core contribution is theoretically sound and intuitive, effectively addressing a known limitation in layer-wise merging strategies (the compounding of representation errors deeper in the network).

2. The empirical validation is thorough, covering large-scale tasks, out-of-domain (OOD) generalization, distribution shifts, and varying data characteristics.

3. The inclusion of a Sequential Merging study is highly appreciated and reflects a realistic deployment scenario for model merging.

**Weaknesses:**

1. Several presentation and writing issues detract from the clarity of the manuscript, particularly in the introduction and motivation sections.

2. The paper lacks a rigorous discussion of why the method underperforms the baseline in specific subsets of the data/layers.

3. Visualizations in the scaling experiments are overcrowded.

**Additional Comments:**

The study on Sequential Merging is a standout addition to this paper. It effectively bridges the gap between static model fusion and continual learning, proving that RegMean++ mitigates forgetting better than standard RegMean as the sequence of tasks grows.

Regarding the findings in Section 5.6 (where standard weight averaging is applied to early layers and RegMean++ to middle/deep layers without significant performance loss): this is a highly practical insight given the 2x computational overhead noted in the Limitations section. Building on this, a highly promising direction for future work would be to develop an algorithm to dynamically identify which layers to merge using RegMean++, and to determine whether other hybrid strategies perform better than simple weight averaging. Automating the discovery of this optimal layer boundary based on task variance, architectural differences, or representation drift would significantly enhance the method's efficiency and scalability.

**References**

Daheim, N., Möllenhoff, T., Ponti, E. M., Gurevych, I., & Khan, M. E. (2024). Model Merging by Uncertainty-Based Gradient Matching. Proceedings of the Twelfth International Conference on Learning Representations (ICLR).

**Audience:**

Yes

**Audience Explanation:**

Model merging is an emerging subfield focused on developing parameter-efficient methods to integrate task-specific knowledge from multiple fine-tuned foundation models, while avoiding the high costs of multi-task retraining. The insights into cross-layer dependency failures present in existing layer-wise merging algorithms will be particularly valuable to researchers involved in model fusion, test-time adaptation, and continual learning.

**Broader Impact Concerns:**

There are no immediate ethical concerns specific to this methodology. The work presents a mathematical and algorithmic improvement to model merging. Standard dual-use concerns regarding the merging of models with harmful capabilities apply, but these do not necessitate a custom broader impact statement beyond standard community guidelines.

**Claims And Evidence:**

Yes

**Claims Explanation:**

The authors provide comprehensive empirical evidence to support their claims. The evaluation spans multiple architectures (ViT-B/32, ViT-B/16, ViT-L/14, Llama-3.1-8B, Llama-3.2-3B) and benchmarks. The Centered Kernel Alignment (CKA) analysis convincingly backs the claim that RegMean++ better aligns latent representations across layers compared to the baseline. Furthermore, the authors transparently report both the accuracy metrics and the computational time/memory requirements.

However, the discussion of negative results is incomplete. The authors show that RegMean++ underperforms RegMean in certain earlier layers (Figure 3: SVHN, Cars, SUN397) and in specific data scenarios (Table 5), but they fail to analyze or hypothesize why this degradation occurs.

**Requested Changes:**

Critical to securing a recommendation for acceptance:

1. Writing and Presentation: The introduction and motivation require proofreading, as several paragraphs are awkwardly phrased. For example, the introduction contains grammatical issues that impede reading, such as: "However, we discuss an important caveat of RegMean is that it operates..." Please revise for academic clarity.

2. Analysis of Negative Results: Address the performance drops observed in earlier layers in Figure 3 (specifically for SVHN, Cars, and SUN397) and in Table 5. Provide a theoretical or empirical discussion on why capturing cross-layer dependencies hurts performance in these specific scenarios.

**Would strengthen the work:**

3. Computational Complexity Discussion: Expand the discussion of the algorithm's computational complexity in terms of the dimensionality of the layers, in order to inform on possible bottlenecks.

4. Additional Baseline (Fisher Merging): The current Fisher Merging baseline is standard, but the state-of-the-art for Hessian/Fisher-based merging has advanced. I suggest including a comparison against (or at least a discussion of) the uncertainty-based gradient matching corrections applied to Fisher Merging, as detailed in Model Merging by Uncertainty-Based Gradient Matching (Daheim et al., ICLR 2024). This would strengthen the positioning of RegMean++ against optimized second-order merging schemes.

5. Figure 5 Accessibility: The plots in Figure 5 are overcrowded and difficult to parse. Please use different line styles, distinct markers, and potentially a larger format to differentiate the algorithms and improve accessibility.

6. Figure 3 Clarity: Since space permits, write out "Multi-Task Learning" instead of using the acronym "MTL" in the legend/labels of the figure to improve standalone readability.

---

> ### Author Response · Authors · 2026-03-23
> **Official Comments by Authors**
>
> We thank the Reviewer for your comments, effort, and time in reviewing our paper. Please find our responses to your concerns below. We have also updated our manuscript; the added contents are highlighted in blue.
>
> ---
>
> > 1. Writing and Presentation: The introduction and motivation require proofreading, as several paragraphs are awkwardly phrased...
>
> **A:** We have updated the introduction and motivation in the revised manuscript accordingly.
>
> > 2. Analysis of Negative Results: Address the performance drops observed in earlier layers in Figure 3 (specifically for SVHN, Cars, and SUN397) and in Table 5. Provide a theoretical or empirical discussion on why capturing cross-layer dependencies hurts performance in these specific scenarios.
>
> **A:** **CKA similarity degradation in early layers.** Different from CNNs, ViTs with attention heads can aggregate global spatial information even at early layers [1]. We conjecture that this effect leads to quantitatively different features learned by early layers of candidate models trained on SVHN, Cars, and SUN397. Because merging results in a single model, incorporating cross-layer dependencies could temporarily induce a slight representation drift in these layers.
>
> **Performance degradation when varying data characteristics.** In Table 5, both RegMean and RegMean++ show significant performance drops as the distribution of the merging data diverges from that of the training data. Although RegMean++ underperforms RegMean when using out-of-distribution ImageNet samples for merging ViT-B/32 models, we conjecture that this degradation is not caused by the cross-layer dependencies but rather the underlying regression mean mechanism. As the merging data diverges further from the training distribution, the calculated statistics become increasingly uninformative, causing the regression mean to fail. Nevertheless, RegMean++ outperforms RegMean in the other cases in Table 5, indicating the advantage of capturing the cross-layer dependencies.
>
> > 3. Computational Complexity Discussion: Expand the discussion of the algorithm's computational complexity in terms of the dimensionality of the layers, in order to inform on possible bottlenecks.
>
> **A:** We have expanded the discussion on computational complexity in terms of dimensionality of the layers for RegMean and RegMean++ in Appendix D.9.
>
> As in Eqn. 2, both RegMean and RegMean++ require matrix inversion to calculate the merged weights. This operation has a time complexity of $O(d^3)$, where $d$ is the dimensionality of the linear layer's weights, making it the primary computational bottleneck. Applying RegMean or RegMean++ to only a subset of transformer layers reduces the total number of matrix inversions, thus decreasing the overall merging time (as detailed in the table below). Furthermore, the time required for these matrix inversions is roughly equivalent to the time required for forward passes.
>
> |Layer subset|ViT-B/32: RegMean|ViT-B/32: RegMean++|ViT-L/14: RegMean|ViT-L/14: RegMean++|
> |:-|:-:|:-:|:-:|:-:|
> |All|31.30|34.17|84.09|171.42|
> |Early|19.70|23.62|36.58|68.69|
> |Middle|20.65|23.58|44.60|88.89|
> |Deep|19.85|26.11|48.63|102.42|
> |Middle \& Deep|22.13|25.74|67.00|134.67|
>
> > 4. Additional Baseline (Fisher Merging): The current Fisher Merging baseline is standard, but the state-of-the-art for Hessian/Fisher-based merging has advanced...
>
> **A:** Thank you for this suggestion. We have added a discussion for this baseline in the Related Works.
>
> > 5. Figure 5 Accessibility: The plots in Figure 5 are overcrowded and difficult to parse...
>
> **A:** Due to the large number of baselines being compared, we found that changing the styles did not work well. Alternatively, we have added Tables 19, 20, 21 with the same results as those in Figure 5 in our revised manuscript.
>
> > 6. Figure 3 Clarity: Since space permits, write out "Multi-Task Learning" instead of using the acronym "MTL" in the legend/labels of the figure to improve standalone readability.
>
> **A:** In our revision, we have updated the legend of Figure 3 (likewise for Figures 4, 12, and 14) accordingly.
>
> > The study on Sequential Merging is a standout addition to this paper...
>
> > Regarding the findings in Section 5.6 (where standard weight averaging is applied to early layers and RegMean++ to middle/deep layers without significant performance loss)...
>
> **A:** We thank the reviewer for highlighting our Sequential Merging study. We appreciate the reviewer's suggestion regarding merging using a subset of layers. We have reflected this suggestion in the Future Work section of our revised manuscript.
>
> [1] Raghu, Maithra, et al. "Do vision transformers see like convolutional neural networks?." Advances in neural information processing systems 34 (2021): 12116-12128
>
> ---
>
> We hope these responses address your concerns. Please take a look at our revised manuscript. If any concerns remain, we are happy to discuss further.

---

### Review · Reviewer_fZ8w · 2026-03-11

**Summary Of Contributions:**

The paper proposes an change to an existing model merging method called RegMean, which the authors call RegMean++.
The authors propose to replace the activations $\mathbf{X}$ that are used to form the objective for RegMean and based on activations of the models that are being merged towards using the activations of the previously merged layers in the merged model.
The authors propose that this accounts for intra-layer and cross-layer dependencies and evaluate the similarity of activations between and within layers. The method is compared against various state-of-the-art methods on a variety of merging benchmarks.

A key weakness in my opinion is that it is unclear whether the "change" of $\mathbf{X}$ actually accounts for intra- and cross-layer dependencies and why the proposed "change" of $\mathbf{X}$ is as well-motivated from a theoretical point-of-view, as elaborated below.
The paper is also hard to read at times and could be presented better, also due to grammar mistakes.

A key strength of the paper is the very extensive experimentation on many benchmarks and against many baselines.
The method provides improvements on some benchmarks but on others (Tab. 6) these are not as clear.
There are also interesting ablations, in particular, on scaling the number of models and how methods generalize to different tasks.

**Additional Comments:**

- How exactly is RegMean privacy preserving? (This claim is made in the introduction)

- The "The" in the title of Section 3 is not needed

- I am not sure if Fig. 2 really "reveals that RegMean leaves important cross-layer and inter-layer interactions unmodeled". Could you please explain this?

- Isn't the superscript in line 4 of Alg. 1 supposed to be (l-1, j)?

- What exactly is gained from the experiments in Sec. 5.6? To me it's unclear why any partial merging should be done, as "all" always performs best?

- Fig. 2: "presentations" -> "representations"
- It's a bit misleading to call the method "training-free" as there is some optimization based on training data (which is needed to calculate \mathbf{X} statistics)

- Calling the method "a generalized extension of RegMean sounds like it is more general than RegMean but that does not seem to be the case.

- Abstract and throughout: merge model -> merged model if it is referring to the model that is the result of merging.

**Audience:**

Yes

**Audience Explanation:**

Yes, model merging as a technique is highly popular and RegMean is often used.

Therefore, new insights and changes to the method would be of interest to at least a sub-community in TMLR.

**Claims And Evidence:**

No

**Claims Explanation:**

- It is not yet clear to me why changing $\mathbf{X}$ is theoretically well-motivated. In RegMean this seems to rather be a matter of convenience, as the $\mathbf{X}$ in the objective (Eq. 1) is also simply seen as given and arbitrary. In RegMean the objective is (implicitly via the algorithm) to make weight matrices of the merged model have the same function output as candidate weight matrices on each candidate weight matrix pre-activation. In RegMean++ the objective is (implicitly via the algorithm) to minimize this function-space discrepancy wrt. pre-activations of the merged model. But why is one intuitively better than the other and why does the latter account for cross- and intra-layer dependencies while the former does not? At first thought it seemed intuitive to me that using the merged model's activations makes sense for the merged model, but then again it does not make as much sense for the candidate models. One might argue that both options for $\mathbf{X}$ are in a way mismatched, in both cases either the merged model or candidate models "see data they do not expect".

- Rather, to me the objective would better be put as minimizing function-space discrepance wrt. an unknown distribution over activations and both methods could in some way be understood as using samples from that space and, instead, it would suggest to use both.

- It is not made clear enough why cross layer similarity is a meaningful statistic to calculate. In Fig. 3 there seems to be no notable difference between RegMean and RegMean++ which also contradicts the claim of the paper that RegMean++ accounts for these cross-layer dependencies, which could be defined more clearly. The authors claim that Fig. 2 "reveals that RegMean leaves important cross-layer and inter-layer interactions unmodeled" but this is not clear.

- There doesn't seem to be a clear benefit on many tasks, for example, the language tasks in Tab. 6

**Requested Changes:**

- This change is crucial for acceptance: it needs to be made clearer why the change of $\mathbf{X}$ in the objective is meaningful and why this is actually an improvement over RegMean.

- This change is crucial to strengthen the work: it needs to be made clearer what cross-layer and intra-layer dependencies are exactly and why the statistics calculated on similarities are useful.

- Strengthens the work: In Sec. 2.2 it would be important to note that the objective is not defined to minimize the prediction differences of the whole network but in terms of the output of the specific linear layer

- Strengthens the work: FIgure 5 needs to be plotted differently as it is very hard to make out differences between methods, similarly for Fig. 6

- Needed for acceptance: There are various grammar mistakes, for example, the last sentence in the first paragraph of the introduction is not complete. Another example is the second to last sentence in the second paragraph of the introduction, where adjectives are listed in place of nouns. These should be revised.

---

> ### Author Response · Authors · 2026-03-23
> **Official Comments by Authors (1/2)**
>
> We thank the Reviewer for your comments, effort, and time in reviewing our paper. Please find our responses to your concerns below. We have also updated our manuscript; the added contents are highlighted in blue.
>
> ---
>
> > it needs to be made clearer why the change of $\mathbf{X}$ in the objective is meaningful and why this is actually an improvement over RegMean.
>
> **A:** During inference, layer $l$ of the merged model receives outputs from layer $l-1$ of the **merged model**, not from the candidate models. Because RegMean optimizes the merged layer $l$ using activations from the candidate models, it introduces a mismatch between the merging and inference times. RegMean++ optimizes the merged layer $l$ using activations produced by the previous merged layer $l - 1$. This directly exposes the data that the merged model will actually see during inference.
>
> > One might argue that both options for $\mathbf{X}$ are in a way mismatched, in both cases either the merged model or candidate models "see data they do not expect".
>
> **A:** We note that, for RegMean++, we only change how activation $\mathbf{X}^{(l)}_i$ is computed, the input feature $\mathbf{X}^{(l, j)}_i$ is computed in the same way as RegMean, *i.e.,* by running a forward pass through a candidate layer. Regarding the reviewer's concern that candidate models may see unexpected data, **this is correct only for the very first linear layers** (*e.g.,* key, query, and value matrices) within the candidate layer. As data flows deeper into the candidate layer, subsequent linear layers (*e.g.,* output, up-projection, and down-projection matrices) see the expected data produced by the preceding components. We have updated Figure 1 and Algorithm 1 in our revised manuscript to better demonstrate this dynamic.
>
> One might argue that if $\mathbf{X}^{(l)}_i$ significantly diverges from the ground-truth distribution, feeding it into the candidate layer might cause significant representation drift for the input features. Consequently, this causes the layer being merged to inherently have a significant representation drift. We show that this is not necessarily true. Our analysis of the cross-layer CKA similarity (Figure 3) indicates that the activations produced by the merged model remain close to those of the candidate models across layer indices and tasks.
>
> > Rather, to me the objective would better be put as minimizing function-space discrepance wrt. an unknown distribution over activations and both methods could in some way be understood as using samples from that space and, instead, it would suggest to use both.
>
> **A:** We studied the effect of using an unknown distribution for RegMean and RegMean++ in Section 5.7. We used ImageNet samples for merging. These samples do not exist in the task-specific training datasets that the candidate models were fine-tuned on. We find that merging using these out-of-distribution samples significantly decreases the performance of both methods.
>
> > it needs to be made clearer what cross-layer and intra-layer dependencies are exactly
>
> **A:** We use these terms to specifically describe the information flow for the **merged model**. Cross-layer dependency refers to the sequential flow of activations *between* consecutive transformer layers. Intra-layer dependency refers to the sequential flow of intermediate features *within* a single transformer layer.
>
> > why does the latter (RegMean++) account for cross- and intra-layer dependencies while the former (RegMean) does not?
>
> **A:** During the merging process, RegMean++ accounts for cross- and intra-layer dependencies by incorporating information flow and feature transformations occurring across and within the merged model's layers. On the other hand, RegMean treats the merged model independently. Intuitively, it does not account for these dependencies of the merged model.
>
> > why the statistics calculated on similarities are useful.
>
> **A:** Cross-layer and intra-layer similarities are useful for comparing the behaviors of the merged model against the candidate models. By observing how CKA similarities change across transformer layers or components within a transformer layer, we can analyze how the behaviors of the merged model change as data propagates deeper into the network. We utilize CKA as it is one of the standard similarity indexes for identifying correspondences between representations of different neural networks [1, 2, 3, 4].

---

> > ### Author Response · Authors · 2026-03-23
> > **Official Comments by Authors (2/2)**
> >
> > > In Fig. 3 there seems to be no notable difference between RegMean and RegMean++ which also contradicts the claim of the paper that RegMean++ accounts for these cross-layer dependencies, which could be defined more clearly.
> >
> > **A:** The supporting evidence for our claim lies in the trajectory of CKA similarity as data flows deeper into the merged model. In most cases, RegMean++ demonstrates improved CKA in the deeper layers. This is particularly evident on EuroSAT, where RegMean++ not only outperforms RegMean but also aligns with the multi-task learning baseline. This confirms that incorporating cross-layer dependencies of the merged model into the merging objective can reduce the representation errors accumulated across layers.
> >
> > > FIgure 5 needs to be plotted differently as it is very hard to make out differences between methods, similarly for Fig. 6
> >
> > **A:** To thoroughly address this issue, we have supplemented these figures with corresponding tables in our revised manuscript. Tables 19, 20, and 21 detail the results for Figure 5; Tables 22, 23, and 24 detail the results for Figure 6.
> >
> > > There doesn't seem to be a clear benefit on many tasks, for example, the language tasks in Tab. 6
> >
> > **A:** We acknowledge this. While RegMean++ consistently improves performance over RegMean across many vision tasks, its performance on language tasks is mixed.
> >
> > > The authors claim that Fig. 2 "reveals that RegMean leaves important cross-layer and inter-layer interactions unmodeled" but this is not clear.
> >
> > **A:** We acknowledge that this claim is an overstatement. Figure 2 shows that CKA similarity decreases in deeper layers due to the accumulation of representation errors throughout the network. It does not reveal that RegMean leaves cross-layer and intra-layer interactions of the merged model unmodeled (please note that "intra-layer" was mistakenly written as "inter-layer" in our initial manuscript). We have removed this from our revised manuscript.
> >
> > > How exactly is RegMean privacy preserving?
> >
> > **A:** By "privacy preserving", we specifically mean "training data privacy preserving". As stated in Section 3.4 of RegMean's original work, RegMean requires only the inner-product matrices when merging and does not require the training data.
> >
> > > What exactly is gained from the experiments in Sec. 5.6?
> >
> > **A:** Section 5.6 studies which components/transformer layers contribute more to merging performance to inform future work on developing efficient merging frameworks. We find that, for example, merging using only the middle and deep layers preserves over 98\% accuracy while reducing the computational overhead (detailed in Appendix D.9 on the bottleneck of RegMean and RegMean++).
> >
> > > Isn't the superscript in line 4 of Alg. 1 supposed to be (l-1, j)?
> >
> > **A:** The superscript of $\(l, j\)$ is correct. As if $\mathbf{X}_i^{(l, j)}$ is the activation for the current merged layer $l$, it is calculated by the previous merged layer $f_M^{(l-1)}$. Nevertheless, we have updated the details of Algorithm 1 in our revised manuscript.
> >
> > > In Sec. 2.2 it would be important to note that the objective is not defined to minimize the prediction differences of the whole network but in terms of the output of the specific linear layer
> >
> > > There are various grammar mistakes, for example, the last sentence in the first paragraph of the introduction is not complete.
> >
> > > The "The" in the title of Section 3 is not needed
> >
> > > Fig. 2: "presentations" -> "representations"
> >
> > > Calling the method "a generalized extension of RegMean sounds like it is more general than RegMean but that does not seem to be the case.
> >
> > > Abstract and throughout: merge model -> merged model if it is referring to the model that is the result of merging.
> >
> > **A:** Thank you for pointing them out. We have fixed these issues in our revised manuscript accordingly.
> >
> > [1] Kornblith, Simon, et al. "Similarity of neural network representations revisited." International conference on machine learning. PMlR, 2019.
> >
> > [2] Nguyen, Thao, Maithra Raghu, and Simon Kornblith. "Do Wide and Deep Networks Learn the Same Things? Uncovering How Neural Network Representations Vary with Width and Depth." International Conference on Learning Representations.
> >
> > [3] Raghu, Maithra, et al. "Do vision transformers see like convolutional neural networks?." Advances in neural information processing systems 34 (2021): 12116-12128.
> >
> > [4] Kim, Dongwan, and Bohyung Han. "On the stability-plasticity dilemma of class-incremental learning." Proceedings of the IEEE/CVF conference on computer vision and pattern recognition. 2023.
> >
> > ---
> >
> > We hope these responses address your concerns. Please take a look at our revised manuscript. If any concerns remain, we are happy to discuss further.

---

### Author Response · Authors · 2026-03-23
**General Response: Additional Results on Gemma 2 Models**

Dear Reviewers,

We have conducted **additional experiments on language tasks** with Gemma-2-2B and Gemma-2-9B as follows.

We show the performance comparison between RegMean++ and baselines for two Gemma 2 variants in the tables below. Similar to the results from Llama 3 variants, the effectiveness of RegMean++ over RegMean is dependent on the scale of the base model: RegMean++ outperforms RegMean on the Gemma-2-2B model (33.0 vs. 32.9), but it underperforms RegMean on the Gemma-2-9B model (50.7 vs. 51.5). Notably, on the Gemma-2-9B model, both RegMean and RegMean++ underperform the data-free baselines (Model Soups, Task Arithmetic, and TIES-Merging). Model Soups achieves the best overall result on the Gemma-2-2B model (33.4), and dominates in the instruction following (18.1), mathematics (33.1), and multilingual (47.9) domains. Task Arithmetic achieves the best overall result on the Gemma-2-9B model (53.9).

*Gemma-2-2B:*

|Method|Instruction following|Math|Multilingual|Coding|Safety|**Avg.**|
|:---|:---:|:---:|:---:|:---:|:---:|:---:|
|Model Soups|**18.1**|**33.1**|**47.9**|30.1|37.6|**33.4**|
|Task Arithmetic|14.2|30.1|46.5|27.9|**40.9**|31.9|
|TIES-Merging|13.7|29.5|45.2|26.3|39.9|30.9|
|RegMean|16.3|31.2|47.1|**32.0**|38.0|32.9|
|**RegMean++**(Ours)|15.9|32.8|46.9|31.5|38.1|33.0|

*Gemma-2-9B:*

|Method|Instruction following|Math|Multilingual|Coding|Safety|**Avg.**|
|:---|:---:|:---:|:---:|:---:|:---:|:---:|
|Model Soups|26.1|68.0|60.0|51.9|62.0|53.6|
|Task Arithmetic|**26.6**|68.1|60.0|51.9|**63.1**|**53.9**|
|TIES-Merging|21.6|66.6|**60.8**|50.8|62.1|52.4|
|RegMean|25.1|**71.5**|58.9|**54.6**|47.5|51.5|
|**RegMean++**(Ours)|22.9|68.8|58.5|53.7|49.6|50.7|

---

### Decision · Action_Editor_kDe2 · 2026-04-21

**Recommendation:** Accept as is

**Audience:**

Yes

**Audience Explanation:**

The paper is about model merging which is a relevant topic for TMLR's audience.

**Claims And Evidence:**

Yes

**Claims Explanation:**

This paper is focuses on improving RegMean. After the rebuttal, two reviewers are mildly leaning accept and one learning reject. The reviewer in favor of rejection agree that the experiments are thorough and have been executed well but they believe that overall the method is a small change to an existing method. Out of the reviewer who lean accept, one reviewer finds that the proposed method reduces "the accumulation of error on internal activations of the merged model" which results in improved performance but the method only seem to have an edge in the Vision domain. The other reviewer also believes that this work is "an okay-ish follow-up on RegMean" with "clear but incremental contributions."

My decision to accept this paper is based on the recommendation of the two reviewer who believe that this work can still be accepted as is and could be useful for the community.

---

> ### Author Response · Authors · 2026-04-25
> **Submission of Camera-ready Version**
>
> Dear Action Editor and Reviewers,
>
> Thank you for your time and effort in organizing and reviewing our paper.
>
> We have submitted the camera-ready version of our submission and the code.
>
> Once again, thank you for your constructive comments throughout the review process.
>
> Best regards,
>
> The Authors